# A sphingolipid-dependent diffusion barrier confines ER stress to the yeast mother cell

**Lori Clay[1][†], Fabrice Caudron[1][†], Annina Denoth-Lippuner[1], Barbara Boettcher[1], Stéphanie Buvelot Frei[1], Erik Lee Snapp[2], Yves Barral[1]***

[1]Institute of Biochemistry, Department of Biology, ETH Zürich, Zürich, Switzerland; [2]Department of Anatomy and Structural Biology, Albert Einstein College of Medicine of Yeshiva University, New York, United States

**Abstract** In many cell types, lateral diffusion barriers compartmentalize the plasma membrane and, at least in budding yeast, the endoplasmic reticulum (ER). However, the molecular nature of these barriers, their mode of action and their cellular functions are unclear. Here, we show that misfolded proteins of the ER remain confined into the mother compartment of budding yeast cells. Confinement required the formation of a lateral diffusion barrier in the form of a distinct domain of the ER-membrane at the bud neck, in a septin-, Bud1 GTPase- and sphingolipid-dependent manner. The sphingolipids, but not Bud1, also contributed to barrier formation in the outer membrane of the dividing nucleus. Barrier-dependent confinement of ER stress into the mother cell promoted aging. Together, our data clarify the physical nature of lateral diffusion barriers in the ER and establish the role of such barriers in the asymmetric segregation of proteotoxic misfolded proteins during cell division and aging.

*For correspondence: yves. barral@bc.biol.ethz.ch

[†]These authors contributed equally to this work

**Competing interests:** The authors declare that no competing interests exist.

**Reviewing editor**: W James Nelson, Stanford University, United States

## Introduction

Cellular diversity in eukaryotes relies in many instances on the asymmetric partition of cell fate determinants between daughter cells at cell division. In such instances, asymmetric cell division generates daughters with different division potentials. For example, most somatic cells exhibit a limited division potential, whereas the stem cells that generate them can divide indefinitely. Understanding the basis for these differences is critical for cell biology, aging research and regenerative medicine.

The yeast *Saccharomyces cerevisiae* divides in an asymmetric manner through the budding of daughters from the surface of the mother cell. While these daughters are born young and form eternal lineages, the mother cells divides only a limited number of times (20–50) before stopping and dying. This process, termed replicative aging (*Egilmez and Jazwinski, 1989*; *Kennedy et al., 1994*; *Steinkraus et al., 2008*), is a consequence of the retention and accumulation of aging factors in the mother cell. A large variety of cellular features have been implicated in limiting the life span of yeast mother cells, including DNA-repair by-products called extra-chromosomal ribosomal DNA circles (ERCs), carbonylated proteins, oxidized lipids (*Nyström, 2005*; *Steinkraus et al., 2008*), multi drug transporters (*Eldakak et al., 2010*), vacuolar pH and mitochondrial integrity (*Hughes and Gottschling, 2012*). How many more factors contribute to aging, whether and how these factors influence each other, which of them are early and primary causes of aging, and which of them actually kill the cell at the end of its life remain unclear. We also know little about how the segregation of these factors is biased towards the mother cell during mitosis.

Recent data indicated that a lateral diffusion barrier in the outer nuclear membrane compartmentalizes the dividing nucleus and promotes the retention of DNA circles in the mother compartment

**eLife digest** Cell division isn't always about splitting a cell into two identical parts. The diversity of many of our own cells relies on asymmetric cell divisions. The yeast used to make bread rely on a process called 'budding' that involves a small daughter cell emerging from the surface of the mother cell. Mother cells can only produce around 20–50 daughter cells before dying from old age. However, their daughters are always born rejuvenated, and not aged like their mothers.

Budding involves part of the plasma membrane that surrounds the mother cell being pinched off to produce the daughter cell. This part of the membrane contains diffusion barriers that prevent various factors—including factors that cause aging—from entering the daughter cell. The barriers are known to contain several layers, but the details of how they work were not understood.

Inside the budding cell, the membrane of the endoplasmic reticulum (ER) also contains lateral diffusion barriers. The ER is the structure in the cell responsible for folding newly made proteins correctly. Any misfolded, toxic proteins are kept in the ER to be refolded or destroyed. However, if there are too many misfolded proteins, the ER gets stressed and triggers a mechanism that in extreme cases causes the cell to self-destruct.

Clay, Caudron et al. have now shown that ER stress causes yeast cells to age. Moreover, when the ER is stressed, the ER diffusion barrier prevents the stress that causes aging entering the daughter cells.

Clay, Caudron et al. also established that the diffusion barrier in the ER is made up of three layers. A layer of fatty molecules called sphingolipids is found at the bottom of the barrier, and such a layer is also present in other diffusion barriers. This could therefore act as the skeleton on which diffusion barriers form. Further investigation of this layer should provide a better understanding of how diffusion barriers work.

(*Shcheprova et al., 2008*) and ERC accumulation (*Lindstrom et al., 2011*). Accordingly, barrier defective cells are long-lived while their successive daughters become progressively shorter lived as they are born to mothers of increasing age. However, these mothers still age, indicating that they still accumulate some aging factors. Furthermore, the retention of old multi drug transporters in the mother cell is independent of the diffusion barriers (*Eldakak et al., 2010*). Thus, several mechanisms control the segregation of aging factors towards the mother cell. However, what these mechanisms are and what their respective contribution to age segregation is remain unclear.

Lateral diffusion barriers have been described in a number of eukaryotic membranes, including the initial segment of axons, dendritic spines, tight junctions of epithelial cells, the base of primary cilia, and the neck of budding yeast cells (*Myles et al., 1984*; *Winckler and Mellman, 1999*; *Barral et al., 2000*; *Takizawa et al., 2000*; *Matter and Balda, 2003*; *Nakada et al., 2003*; *Luedeke et al., 2005*; *Vieira et al., 2006*; *Shcheprova et al., 2008*; *Caudron and Barral, 2009*). However, we still know very little about their physical nature and their mechanisms of action. The membrane systems of budding yeast cells are compartmentalized by at least three lateral diffusion barriers, one in the plasma membrane (*Barral et al., 2000*; *Takizawa et al., 2000*), one in the cortical ER (cER, *Luedeke et al., 2005*) and one in the outer membrane of the dividing nucleus (*Shcheprova et al., 2008*; *Boettcher et al., 2012*). Their assembly at the bud neck depends on a family of filament-forming GTPases, the septins (*Faty et al., 2002*; *Weirich et al., 2008*; *Hu et al., 2010*; *Kim et al., 2010*; *Saarikangas and Barral, 2011*), and on the actin- and formin-interacting protein Bud6 (*Amberg et al., 1995*, *1997*; *Luedeke et al., 2005*; *Shcheprova et al., 2008*). Numerous questions remain concerning their molecular composition, their assembly, and their respective roles in cellular physiology.

The ER is the site of folding and maturation of secretory proteins and protein complexes. A significant fraction of nascent secretory proteins fail to fold, are not correctly glycosylated, or are unable to find their destined partners (*Turner and Varshavsky, 2000*; *Ellgaard and Helenius, 2003*; *Princiotta et al., 2003*). When accumulating, these misfolded proteins activate the unfolded protein response (UPR) and are recognized by the ER-associated degradation (ERAD) machinery, retrotranslocated to the cytoplasm, polyubiquitinated and targeted for degradation by the 26S proteasome (*Turner and Varshavsky, 2000*; *Ron and Walter, 2007*; *Brodsky and Skach, 2011*). These quality control pathways play an important role in preventing or responding to ER stress, which can otherwise lead to cell death (*Tabas and Ron, 2011*). However, whether ER stress contributes to aging is unknown.

To address the nature and function of the different diffusion barriers in the yeast ER, we investigated the mechanisms underlying the compartmentalization of the cortical ER and their potential contribution to the retention of misfolded ER proteins in the mother cell.

## Results

### Misfolded ER-proteins are retained in yeast mother cells

To determine how misfolded secretory proteins distribute between mother and bud during yeast cell division, we detected them using a recently developed live cell microscopy assay based on fluorescence recovery upon photobleaching (FRAP). Briefly, the endogenous copy of the ER Hsp70 chaperone BiP/Kar2 is expressed as a fusion protein with the super-folding green fluorescent protein (sfGFP) to form Kar2-sfGFP. The sfGFP is followed by the amino acids HDEL which are the yeast equivalent of the ER retention signal (KDEL in animal cells), such as to maintain it in the ER-lumen. This protein is fully functional as a chaperone (*Lajoie et al., 2012*). The diffusion of this Kar2-sfGFP, as recorded by FRAP, slows down as the levels of unfolded and aggregated proteins, with which it interacts, increase (*Lajoie et al., 2012*). We exploited this assay to investigate the occupancy of Kar2-sfGFP with clients in the mother and bud compartments. In homeostatic wild type cells, the initial half time ($t_{1/2}$) of Kar2-sfGFP fluorescence recovery was 8.5 ± 2.3 s in the mother compartment and similarly, 7.7 ± 1.9 s in the bud (*Figure 1A,C*). Consistent with the previous reports (*Lajoie et al., 2012*), the mobility of Kar2-sfGFP dramatically decreased upon treating cells with a lethal dose of tunicamycin (Tm, 1 µg/ml), an inhibitor of N-linked glycosylation, in both mother and bud ($t_{1/2}$ = 25.6 ± 9.8 s and 23.4 ± 10.1 s, respectively, after 2 hr, data not shown). However, when cells where treated with a non-lethal dose of the drug but

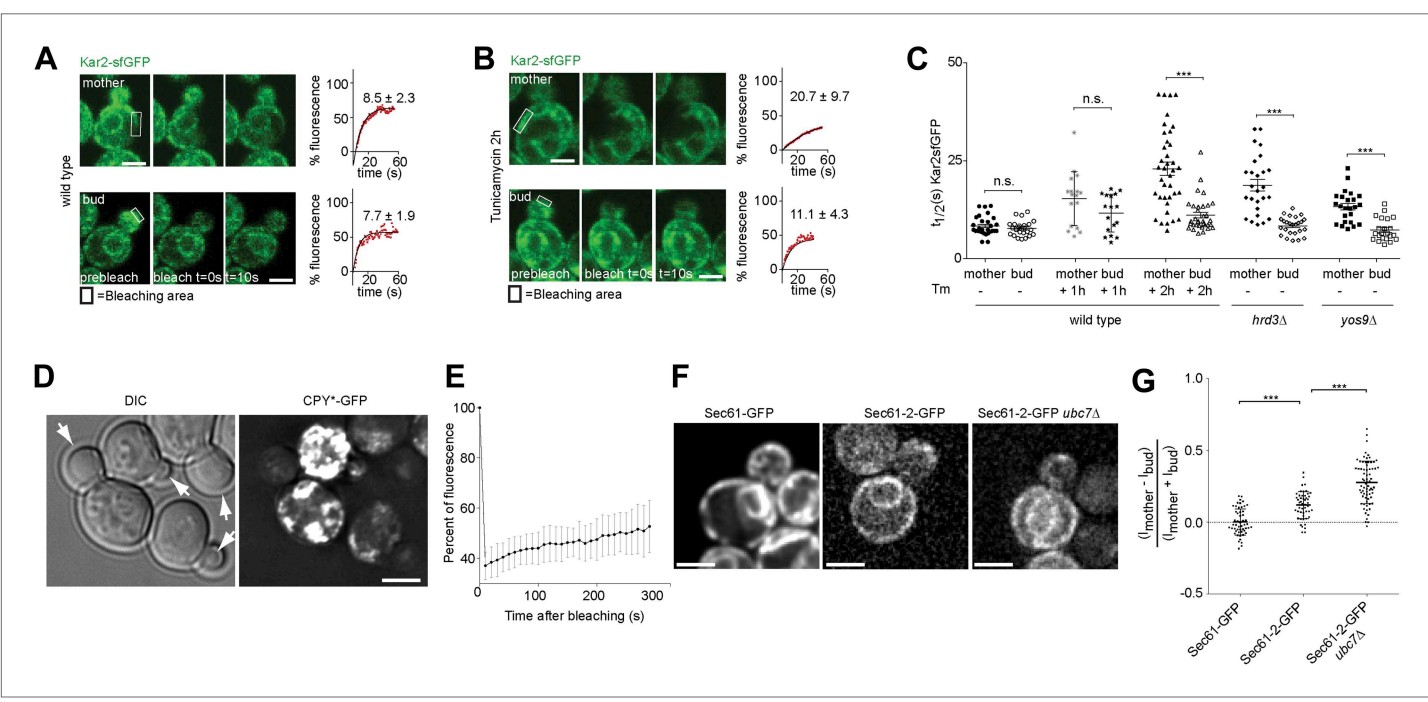

**Figure 1**. Misfolded proteins are retained in the mother cell during budding yeast division. (**A**) FRAP analysis of Kar2-sfGFP in wild type mothers (n = 29 cells) and buds (n = 24 cells) under normal growth condition and (**B**) in the presence of 0.5 µg/ml of Tm. Representative cells are shown. Rectangles indicate the bleached areas. Image show cell before bleaching (pre-bleach), immediately after bleaching (0 s) and 10 s after the initial bleach (10 s). $T_{1/2}$ represents the time it took to recover 50% of fluorescence of the reached plateau. Graphs of fluorescence recovery of the corresponding cells are shown. (**C**) Graph indicates the distribution of $t_{1/2}$ for individual cells of each mutant tested in the presence (+) or absence (−) of 0.5 µg/ml Tm) n >25 cells. n.s. = not significantly different ***p<0.001 (*t* test). (**D**) Z-stack projections of CPY*-GFP and corresponding DIC images in wild type cells are shown. More images are shown in Figure 6. (**E**) Quantification of the FRAP of CPY*-GFP. (**F**) Single focal plane of Sec61-GFP and Sec61-2-GFP in wild type or *ubc7Δ* mutant cells. Note that the gray values have been scaled such as to be able to see both proteins. Otherwise, Sec61-2-GFP values are 5–8-fold lower than wild type Sec61-GFP. (**G**) Distribution of asymmetry indices of Sec61-GFP and Sec61-2-GFP in wild type or *ubc7Δ* mutant cells ***p<0.0001 (*t* test). Average ± SD are indicated (**A**, **B**, **C**, **E**, **G**). Scale bars = 2 µm.

for a longer time (0.5 µg/ml for 2 hr), to let misfolded proteins progressively accumulate, the situation was quite different (*Figure 1B*). Kar2-sfGFP diffusion slowed down with the incubation time in tunicamycin ($t_{1/2}$ = 15.1 ± 6.8 after 1 hr and $t_{1/2}$ = 20.7 ± 9.7 s after 2 hr, *Figure 1C*). The fraction of fast particles derived from the FRAP curves dropped from 65.7% in untreated mother cells to 19.0% in mother cells treated for 2 hr with Tm. Whereas Kar2-sfGFP diffusion was clearly slowed down in the mother cell, it remained quick in the bud ($t_{1/2}$ = 11.4 ± 4.8 s after 1 hr and $t_{1/2}$ = 11.1 ± 4.3 s after 2 hr, *Figure 1C*), indicating that misfolded proteins accumulated slowly in the mother cell and did not invade the bud.

Similar results were obtained when we performed the same assay in cells lacking either the ER quality control lectin Yos9 or its binding partner, the membrane protein Hrd3. $T_{1/2}$ in the mother part of *yos9Δ* mutant cells was 13.3 ± 4 s and 18.8 ± 7.4 s in *hrd3Δ* mutant cells (*Figure 1C*), showing that Kar2-sfGFP dynamics were significantly slower than in the mother part of unstressed wild type cells ($t_{1/2}$ = 8.1 ± 2.2 s). As observed with tunicamycin (Tm), $t_{1/2}$ of Kar2-sfGFP was comparable in the buds of the *yos9Δ* (7.3 ± 2.9 s) and *hrd3Δ* mutant cells (8.4 ± 2.3 s) and those of untreated wild type cells (8.0 ± 2.0 s, *Figure 1C*). Therefore, impairing protein quality control in the ER by deleting *YOS9* or *HRD3* leads to a measurable increase in misfolded proteins in the mother part of the cell over time, but little in the bud. Since the *yos9Δ* and *hrd3Δ* mutant cells are also defective in ER associated degradation of misfolded proteins (ERAD), this data excludes the possibility that the low abundance of misfolded proteins in the bud compared to the mother is due to a more potent ERAD pathway in the bud.

Since the data above suggested that misfolded proteins are retained in the mother part of budded yeast cells, we wanted to test this possibility more directly through visualization of such proteins. Thus, we next characterized the localization of CPY*, a mutated form of the vacuolar carboxypeptidase Y (CPY) that fails to properly fold and exit the ER. For visualization, we overexpressed GFP-tagged CPY* using the *GAL1*-promoter and monitored the fate of the protein by time-lapse microscopy. Over time, CPY*-GFP accumulated up to high levels in mother cell, but not in their buds. Even cells that underwent budding with high levels of CPY*-GFP in the ER in the mother cell showed no or very low levels of GFP signal in their growing bud (*Figure 1D*, see also Figure 6C, upper panel). Prolonged overexpression of CPY*-GFP lead to the formation of strongly fluorescent dots in the mother cell that progressively became static. In order to monitor the dynamics of these aggregates, we performed a FRAP experiment on these cells. The aggregates did recover some fluorescence (from 37% ± 5.4% 10s after bleaching to 53% ± 10.4% 5 min after bleaching, n = 10 cells, *Figure 1E*). Therefore, the dynamics of the aggregates themselves is rather slow. Following the cells that underwent budding, we observed that the CPY*-GFP dots remained in the mother cell (Figure 6C). The buds started to form dots themselves only after they had separated from their mother. Thus, no aggregate or otherwise misfolded material appeared to diffuse from the mother cell into its bud.

As a second reporter, we chose the mutant protein Sec61-2, a resident ER protein inserted in the ER-membrane. At high temperature, Sec61-2 misfolds and is rapidly degraded. We tagged this allele with GFP and followed its localization. Its levels were already low at permissive temperature, and rapidly decreased upon incubation at 37°C as already shown for the untagged mutant protein (*Bordallo et al., 1998*). We rationalized that even at permissive temperature, Sec61-2 is significantly misfolded. Therefore, we compared the segregation of Sec61-2 protein to the wild type, correctly folded, Sec61 protein in cells grown at 30°C (*Figure 1F*). We measured the cortical ER mean fluorescence intensity in the mother cell (Im) and its growing bud (Ib), choosing the frame before anaphase as a reference point and calculated its asymmetry index ((Im−Ib)/(Im + Ib)). This index had a value of 0.00 ± 0.09 (n = 52 cells) for wild type Sec61-GFP showing that Sec61-GFP distributes equally between the cortical ER of the mother cell and its bud (*Figure 1G*). However, in cells expressing Sec61-2-GFP the asymmetry index reached a significantly higher value (0.12 ± 0.09, n = 49 cells; p<0.0001 *t* test, *Figure 1F–G*). We then used a strain in which Sec61-2-GFP is a bit more stable, since the ERAD gene *UBC7* is deleted. In *ubc7Δ* mutant cells the asymmetry index of Sec61-2-GFP reached an even higher value (0.27 ± 0.14, n = 71 cells) than in the single Sec61-2-GFP mutant cells (*Figure 1F,G*, p<0.0001 *t* test), consistent with a stabilization of misfolded Sec61-2 proteins and their specific retention in the mother cell. Unlike for CPY*-GFP, no large dots or aggregates were observed. Therefore, we conclude that misfolded Sec61-2 is asymmetrically retained in the mother compartment, and that retention does not require aggregation.

Collectively, these results indicate that misfolded proteins are indeed not shared equally between mother and bud, but retained in the mother cell.

## The ER diffusion barrier requires sphingolipids

We next wondered whether retention of misfolded ER-proteins in the mother cell relied on the lateral diffusion barrier in the ER membrane at the bud neck (*Luedeke et al., 2005*). To determine whether confinement of ER stress to the mother cell required ER compartmentalization, we next set out to characterize in more details how the ER diffusion barrier is established. Based on its published genetic interaction with the septin mutation *cdc3-1* (*Costanzo et al., 2010*), we discovered that the sphinganine C4-hydroxylase Sur2 is required for proper barrier formation. Evidence for ER compartmentalization was obtained using fluorescence loss in photobleaching (FLIP) (*Ellenberg et al., 1997*) as previously described (*Luedeke et al., 2005*), using the ER membrane protein Sec61-GFP as a reporter. Upon repeated photobleaching of a small area of the cortical ER in the mother cell, fluorescence loss elsewhere in the cell is proportional to the exchange rate of the reporter protein between the considered and the photobleached area. As reported for other ER-membrane proteins (*Luedeke et al., 2005*), Sec61-GFP fluorescence is rapidly depleted throughout the wild type mother cell (*Figure 2*, red curves), but not in her bud (green curves), indicating that exchange through the bud neck is reduced. In contrast, in the same assay the luminal reporter protein GFP-HDEL is lost with nearly the same

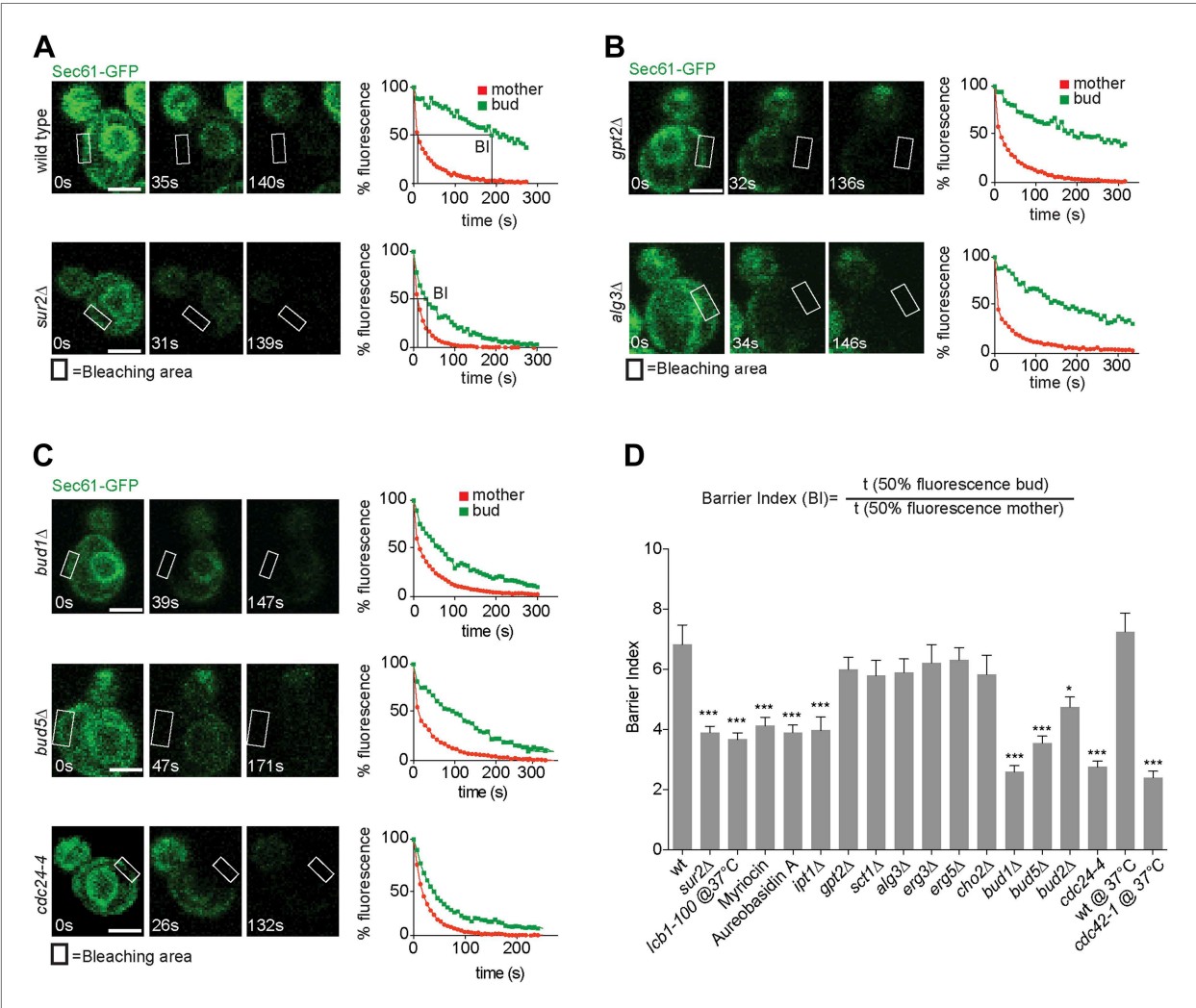

**Figure 2**. Sec61-GFP compartmentalization depends on sphingolipids and Bud1 module. (**A–C**) FLIP of Sec61-GFP in wild type and indicated mutant cells. Fluorescence level decay over time in the mother (red) and daughter part (green) are shown. BI = barrier Index, corresponding to the time of 50% fluorescence loss in the bud divided by the time of 50% fluorescence loss in the mother cell. (**D**) Graph indicates BI + SD of tested strains. n > 20 cells. ***p<0.001, *p<0.05 (*t* test). Scale bars = 2 µm.

kinetics in the bud as in the mother cell (*Luedeke et al., 2005*; *Boettcher et al., 2012*), demonstrating that the barrier is specific for membrane proteins.

To quantify this phenomenon, we used the 'barrier index' (BI, *Luedeke et al., 2005*), defined as the t50 (time to lose 50% of fluorescence) in the bud divided by the t50 in the mother (*Figure 2A,D*). The value of this index increases when exchange between mother and bud becomes limited. In wild type cells, the BI was 7.5 ± 2.3. In contrast, in *sur2Δ* mutant cells Sec61-GFP compartmentalization was reduced, with a BI of 3.5 ± 1.1 (*Figure 2A,D*). Based on this, we rationalized that sphingolipids may contribute to barrier formation. To test this hypothesis, we used the same assay to characterize the involvement of other genes required for sphingolipid biosynthesis. We disrupted the serine palmitoyl-transferase Lcb1 (required for the first step of sphingolipid biosynthesis) by either using a temperature sensitive allele (*lcb1-100*) or the Lcb1 specific inhibitor Myriocin, the ceramide phosphoinositol transferase Aur1 using the inhibitor Aureobasidin A and the Inositolphosphotransferase Ipt1 (involved in the synthesis of mannose-(inositol-P)2-ceramide) through deletion of its gene. In all these cases, Sec61-GFP compartmentalization was strongly affected (BI between 3.6 and 4, *Figure 2D*). In contrast, the ER diffusion barrier was not significantly affected upon inhibition of the glycerolipid biosynthesis pathway (*gpt2Δ, sct1Δ*, BI = 6 and 5.8, respectively, *Figure 2B,D*), the phospholipid biosynthesis pathway (*cho2Δ* BI = 5.8), the lipid-linked oligosaccharide biosynthesis pathway (*alg3Δ* BI = 5.9, *Figure 2B,D*) or the ergosterol biosynthesis pathway (*erg3Δ, erg5Δ* BI = 6.2 and 6.3, respectively, *Figure 2D*). Thus, we concluded that formation of the diffusion barrier in the cortical ER at the bud neck specifically required the sphingolipid biosynthesis pathway to be functional.

## Factors required for bud site selection and polarized growth are also involved in the establishment of the ER diffusion barrier

Previous work on the ER diffusion barrier showed that it is established as soon as the bud emerges (*Luedeke et al., 2005*). It is therefore important that factors required for its establishment localize early on to the bud site. Therefore, we speculated that factors acting in early polarization events, leading to bud emergence, might also promote barrier formation. The earliest event in bud formation consists in the selection of the future site of bud emergence and the recruitment of the polarization machinery. This is achieved by the GTPase Bud1/Rsr1(*Bender and Pringle, 1989*), which is activated at the future site of bud emergence and recruits and activates Cdc24, the guanine-nucleotide exchange factor (GEF) for the Rho-related GTPase Cdc42, a broadly conserved regulator of cellular polarity in eukaryotes (*Chant et al., 1991*; *Powers et al., 1991*; *Bender, 1993*; *Park et al., 1993, 2002*; *Zheng et al., 1995*; *Shimada et al., 2004*). At the plasma membrane, Bud1 is activated by its own GEF, Bud5, which is deposited at the site of the previous cytokinesis through its interaction with the septin cyto-skeleton. Upon bud emergence Bud5 re-localizes to the neck of the newly growing bud, where it stays throughout bud growth and where its function is unknown. The Bud2 protein forms the GTPase acti-vating protein (GAP) involved in Bud1 inactivation. Due to the ability of the Cdc24/Cdc42 module to auto-activate, *bud1Δ* cells still form buds, but at random positions.

To determine whether Bud1 is involved in the formation of the ER diffusion barrier, we assayed Sec61-GFP compartmentalization using FLIP in *bud1Δ*, *bud2Δ* or *bud5Δ* mutant cells (*Figure 2C,D*). Deletion of either of *BUD1* and *BUD5* strongly abolished the compartmentalization of Sec61-GFP (BI *bud1Δ* = 2.4 ± 1.3, *bud5Δ* = 3.5 ± 1.3). The deletion of *BUD2* had a milder effect (BI = 4.8 ± 1.9). Consistent with these results, when Sec61 was tagged with the photoconvertible protein EOS very little of the protein photoconverted in the mother cell diffused into the bud of wild type cells within a 2 min chase (*Figure 3A,B*). In contrast, up to three times more of the protein leaked into the bud already within the first 40 s of the experiment when photoconversion was carried out in the mother of *bud1Δ* mutant cells (*Figure 3A,B*). We concluded that Bud1-GTP contributes to the formation of the diffusion barrier in the ER-membrane at the bud neck cortex.

To test whether Bud1 promoted barrier formation through its interaction with Cdc24 or through another effector, we next investigated whether interrupting the interaction between Bud1 and Cdc24 was sufficient to mimic the effect of Bud1 inactivation. A *CDC24* mutant allele, *cdc24-4*, abolishes this interaction at room temperature (*Shimada et al., 2004*), a condition under which it is otherwise fully functional, that is, promotes normal cellular polarization and bud emergence, albeit at a random pos-ition. Thus, we tested whether the *cdc24-4* mutant cells still compartmentalized the ER membrane properly under this regime. In addition, we used the *cdc42-1* temperature sensitive allele to test whether Cdc42 contributed to barrier formation. FLIP experiments were carried out using Sec61-GFP

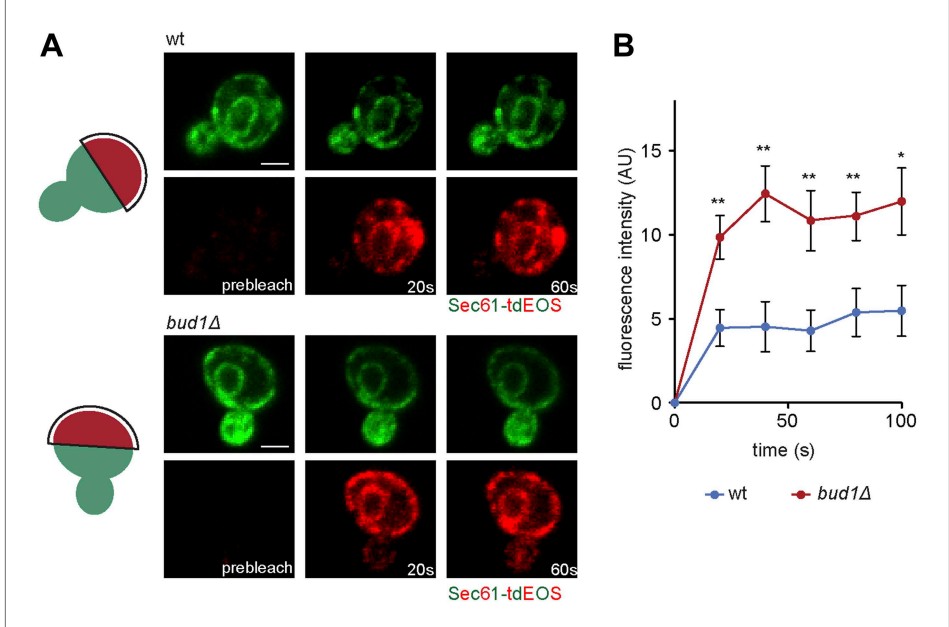

**Figure 3**. Sec61-EOS is compartmentalized in a Bud1-dependent manner. Photoconversion of Sec61-tdEOS in wild type and *bud1Δ* mutant mother cells. (**A**) Images show the red and green fluorescent micrograph before conversion and after 20 s and 60 s. Scheme indicates the converted area. (**B**) Graph of normalized red fluorescence intensity measured in the bud compartment. n > 15 cells, error bars depict the standard error of the mean (SEM). **p<0.01, *p<0.05 (*t* test). Scale bars = 2 μm.

as a reporter in the *cdc24-4* mutant cells at room temperature and in the *cdc42-1* mutant cells shifted to the restrictive temperature (37°C) after bud emergence (*Figure 2C,D*). Both, *cdc24-4* (BI = 2.8 ± 1.1) and *cdc42-1* mutations (BI = 2.5 ± 1.4) abolished barrier assembly in the cortical ER-membrane to the same extent as *BUD1* inactivation. Taken together, our results indicate that Bud1 mediates the formation of the diffusion barrier in the cortical ER essentially through regulation of Cdc24 and activation of the GTPase Cdc42.

## Sphingolipids, but not Bud1, are required for the barrier in the nuclear envelope

After identifying the signaling GTPases Bud1 and Cdc42 and sphingolipids as major players in the assembly of the diffusion barrier in the cortical ER, we next wondered whether these factors generally functioned in the assembly of diffusion barriers. Thus, we asked whether these molecules also contributed to the assembly of the diffusion barrier in the nuclear envelope (*Shcheprova et al., 2008*; *Boettcher et al., 2012*). We performed FLIP on either the nuclear pore complex component Nup49 or the outer nuclear membrane protein Nsg1, each tagged with GFP. For both reporters, repeated photobleaching in a small area of early anaphase nuclei in the mother cell caused the rapid and complete depletion of fluorescence in the entire mother part of the nucleus. In contrast, the fluorescence in the bud part of the anaphase nucleus of wild type cells remained high (BI = 23.2 ± 5.4 for Nup49, *Figure 4A* and 9.5 ± 3.7 for Nsg1, *Figure 4B*) as previously reported (*Shcheprova et al., 2008*). In the *sur2Δ* mutant cells, the fluorescence loss in the bud was significantly faster than in wild type cells (BI = 12 ± 1.4 for Nup49, *Figure 4A* and 4.9 for Nsg1, *Figure 4B*). Remarkably, the *bud1Δ* mutation affected the compartmentalization of neither Nup49 nor Nsg1 (BI = 26.1 ± 13.1 and Nsg1-GFP 10.9 ± 2.6, respectively, *Figure 4A,B*). Similarly, the *cdc24-4* mutation did not affect nuclear envelope compartmentalization at room temperature (BI = 9.1 ± 2.4 for Nsg1, *Figure 4B*). Therefore, the Bud1 signaling module functions in the assembly of the barrier in the cortical ER specifically, whereas sphingolipids are involved in the formation of barriers in both the cortical ER and the nuclear envelope.

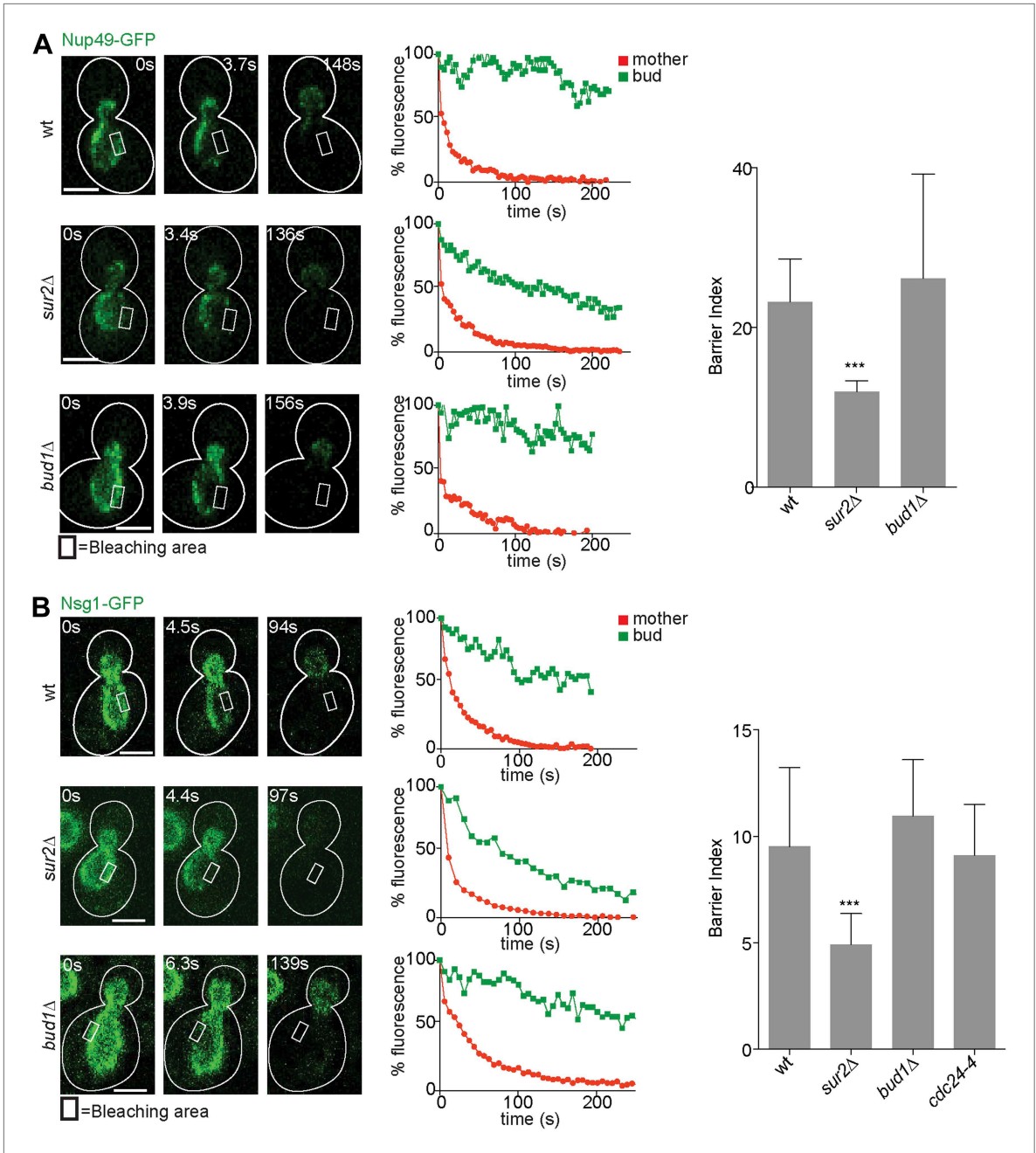

**Figure 4**. Compartmentalization of Nup49-GFP and Nsg1-GFP depends on the sphingolipids, but not on *BUD1*. (**A** and **B**) FLIP experiments and BI values for the indicated markers of the nuclear envelope during early anaphase, in the cells of indicated genotype. Fluorescence level decay over time in the mother (red) and daughter part (green) are shown. White lines indicate cell outlines. Representative experiments are shown. n > 20 cells. ***p<0.001 (*t* test). Scale bars = 2 μm.

## The ER diffusion barrier prevents misfolded proteins from entering the daughter cell

Taking advantage of this information, we asked whether the retention of ER stress in the mother cell required the ER diffusion barrier. In order to determine this, we first analysed the dynamics of Kar2-sfGFP in *sur2Δ* and *bud1Δ* mutant cells. The $t_{1/2}$ of Kar2-sfGFP in mother and buds of these mutant cells was very similar to wild type values (8.2 ± 2.1 s and 7.7 ± 2.6 s in *sur2Δ*; 7.6 ± 2.6 s and 7.4 ± 2.5 s in *bud1Δ* mutant cells; *Figure 5A,B*), indicating that neither of these mutations caused misfolding or

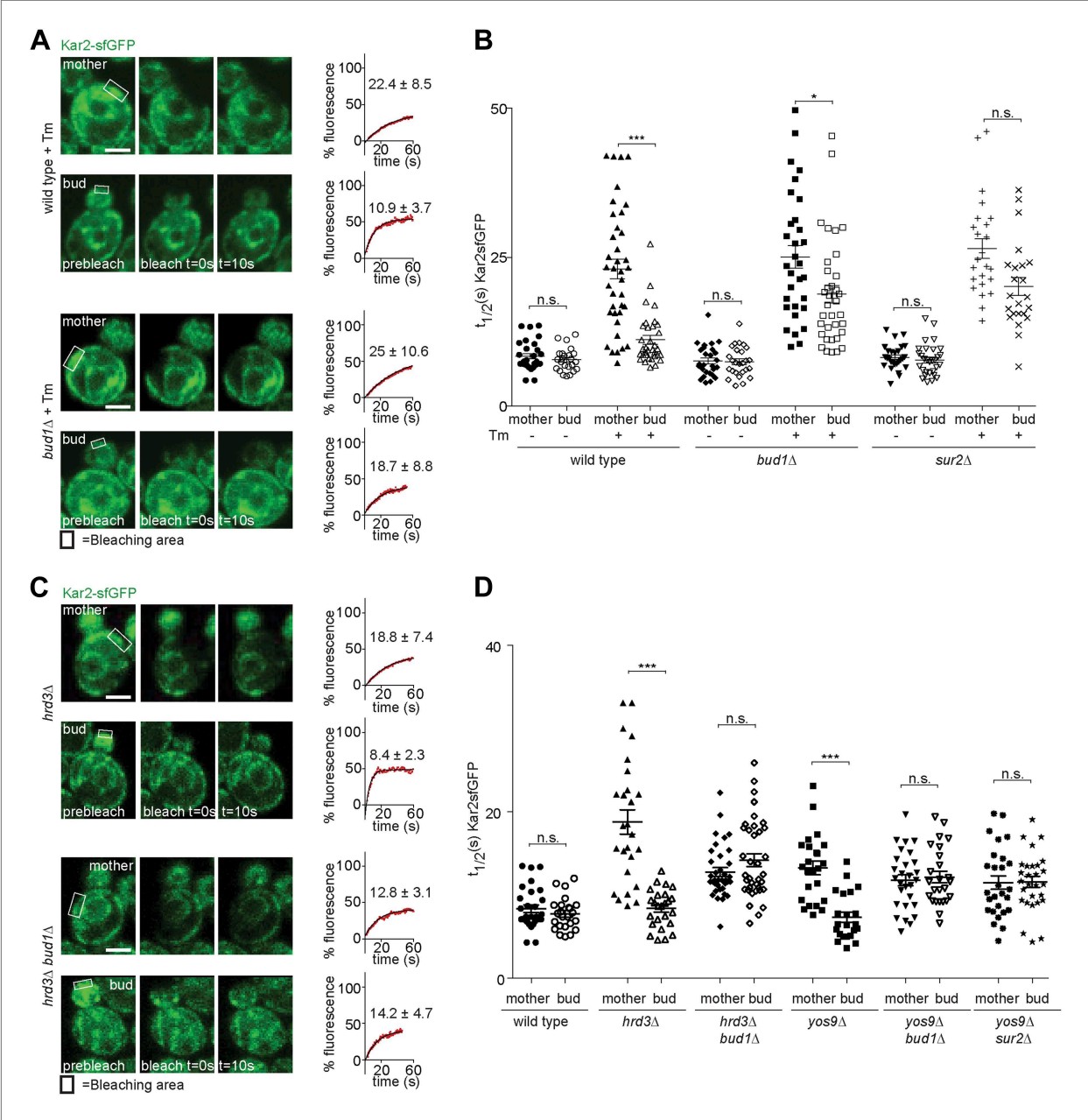

**Figure 5**. Retention of Kar2 clients in the mother cell depends on the ER diffusion barrier. (**A** and **C**) FRAP of Kar2-sfGFP in the mother or bud of mutants of indicated genotype in the presence or absence of 0.5 µg/ml tunicamycin. (**B** and **D**) Graphs indicate the distribution of individual $t_{1/2}$ values measured in mutant cells of indicated genotype. Representative cells are shown. n > 20 cells. Averages ± SD are indicated. n.s. = not significantly different, ***p<0.0001, *p<0.05 (*t* test). Scale bars = 2 µm.

ER stress on their own. As reported earlier, treatment with Tm (2 hr, 0.5 µg/ml) slowed down Kar2 dynamics over time in the mother cell much more than in the daughter cell (**Figure 1B**). Similarly, Kar2-sfGFP mobility strongly decreased in the *sur2Δ* and *bud1Δ* mutant cells treated with Tm (in mother cells $t_{1/2}$ = 26.5 ± 8.1 and 25 ± 10.6, respectively; **Figure 5A,B**). However, the asymmetry of Kar2-sfGFP between mother and bud was lost, the mobility of Kar2-sfGFP being considerably slowed down in the bud as well (*sur2Δ* = 19.7 ± 7.4 and *bud1Δ* =18.7 ± 8.8, **Figure 5A,B**). Thus, inactivation of Sur2 and Bud1 did not affect the appearance of misfolded proteins and the emergence of ER stress in the presence of Tm, but caused misfolded proteins to distribute more equally between mother and bud.

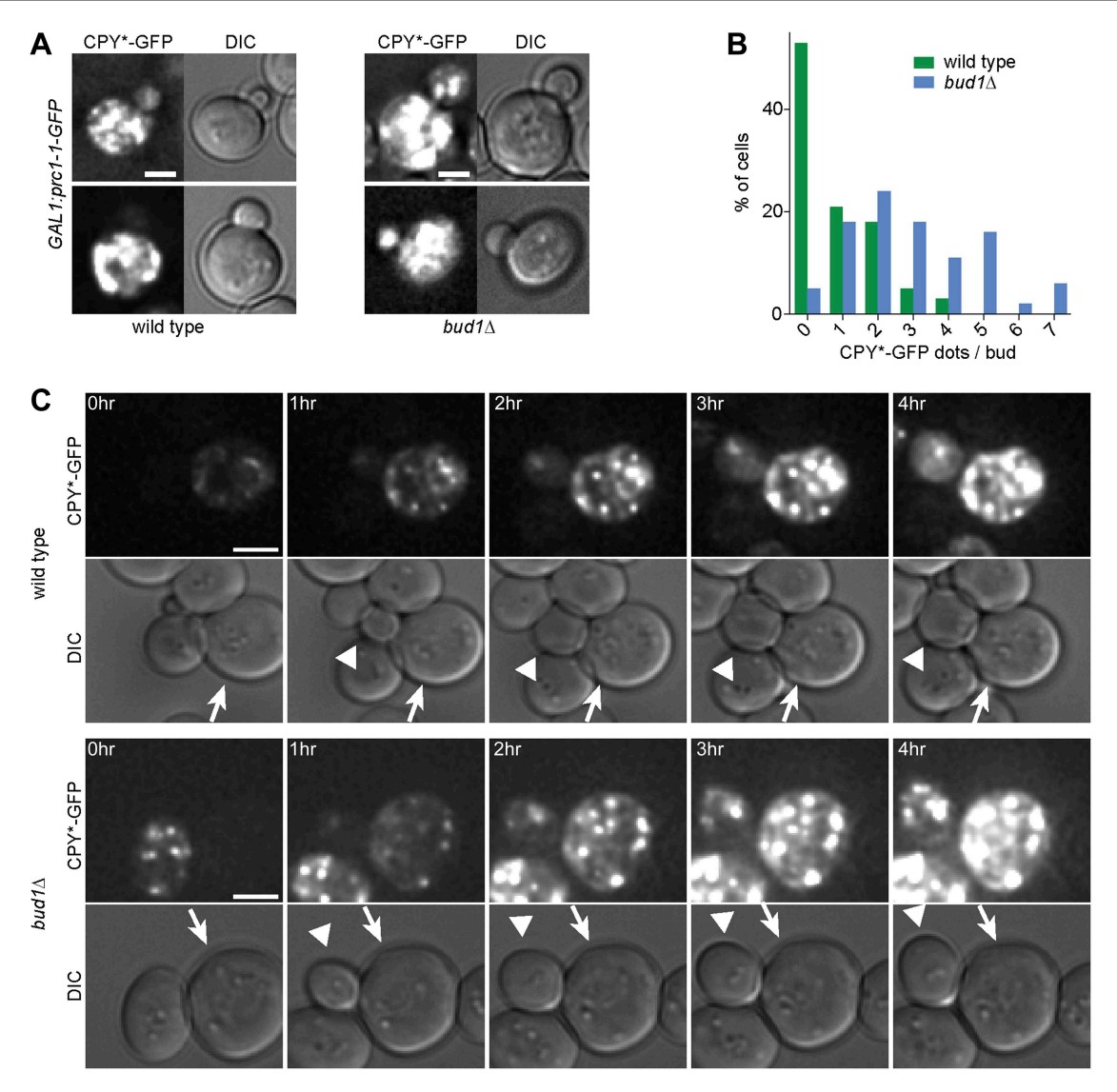

**Figure 6**. The cER diffusion barrier retains misfolded CPY*-GFP in the mother cell. (**A**) Z-stack projections of CPY*-GFP and corresponding DIC images in wild type and *bud1Δ* mutant cells. (**B**) Graph shows the frequency of buds with indicated number of CPY*-GFP dots in wild type and *bud1Δ* mutant cells. (**C**) DIC and CPY*-GFP fluorescence images of time lapse microscopy of wild type (up) and *bud1Δ* mutant cells (bottom). The arrow indicates the mother cell and the arrowhead the growing bud. Scale bars = 2 μm.

Similarly, deleting *SUR2* or *BUD1* in the *yos9Δ* mutant cells lead to the mobility of Kar2-sfGFP to increase symmetrically, that is in both mother and bud ($t_{1/2}$ = 11.5 ± 4.2 and 11.6 ± 3.5 respectively, in the *yos9Δ sur2Δ* double mutant cells; 11.8 ± 3.5 and 12.2 ± 3.5 in the *yos9Δ bud1Δ* double mutant cells, ***Figure 5D***). Inactivation of *BUD1* in the *hrd3Δ* mutant cells had the same effect ($t_{1/2}$ mother: 12.8 ± 3.1, $t_{1/2}$bud: 14.2 ± 4.7, ***Figure 5C,D***).

To test whether weakening of the diffusion barrier affected the distribution of the misfolded protein CPY*-GFP between mother and bud as well, we expressed CPY*-GFP under the *GAL1*-promoter in the *bud1Δ* mutant cells and monitored the number of aggregates present in the buds (***Figure 6A***). In the majority of wild type buds (53%, n = 32 cells), we did not detect any CPY*-GFP aggregates (***Figure 6B***). In contrast, only 5% of *bud1Δ* mutant buds (n = 36 cells) were free of aggregates. More than 85% of the *bud1Δ* mutant buds had 1 to 5 dots, whereas 90% of wild type buds had maximum 2 dots (***Figure 6B***). The increasing number of aggregates in the buds of these barrier-defective cells was not due to the diffusion of aggregates into the bud, since we never observed such an event, but to the fact that the buds

started to form aggregates themselves much before they separated from their mother cell, that is, much earlier than wild type buds (*Figure 6C*). These results indicate that wild type cells retained both CPY*-GFP aggregates and non-aggregated but misfolded CPY*-GFP in the mother cell. Whereas retention of the aggregates was not barrier-dependent, the retention of smaller, not yet aggregated entities was.

We then asked whether the retention of Sec61-2-GFP in the mother cell also depended on an intact diffusion barrier in the cortical ER. We followed single cells expressing Sec61-2-GFP by time-lapse microscopy. Upon *UBC7* deletion Sec61-2-GFP is highly asymmetrically retained in the mother part of the cortical ER (*Figure 1G*, *Figure 7A,B*). This retention was decreased upon the deletion of *BUD1* (asymmetry index of 0.07 ± 0.12, n = 46 *ubc7Δ bud1Δ* Sec61-2-GFP mutant cells, p<0.0001 *t* test compared to *ubc7Δ* Sec61-2-GFP mutant cells, *Figure 7A,B*). Therefore, misfolded Sec61-2-GFP proteins were also retained in the mother cell in a Bud1-dependent manner. Together, our data indicate that aggregation and the compartmentalization of the ER by the diffusion barriers at the bud neck are required for the confinement of misfolded proteins into the mother cell.

## Control of longevity by the cortical ER and nuclear envelope diffusion barriers

The accumulation of misfolded proteins is toxic for the cell. Therefore, we wondered whether a mild ER stress may act as an aging factor. When pedigree analyses were carried out on plates containing very low and non-lethal levels of Tm (0.2 µg/ml), we observed dramatic effects on cellular longevity, which caused most cells to die after 2–4 generations. We therefore decided not to use Tm to study the replicative life span of cells with ER stress. Instead, we performed pedigree analysis of *yos9Δ* mutant cells. A clear shortening of cellular longevity was observed in theses cells (median life span of 19 generations, *Figure 8A*) compared to wild type cells (median life span of 26 generations). Therefore, the accumulation of ER stress in the cells leads to the shortening of their replicative potential and ER stress is indeed an aging factor.

We then asked whether the diffusion barrier contributed to any substantial extent to the accumulation of ER stress in the mother cell with age, and the effect of ER stress on aging. Supporting this view, the *yos9Δ bud1Δ* double mutant cells (22 generations) lived similarly long as the *bud1Δ* (23 generations) and significantly longer than the *yos9Δ* single mutant cells (*Figure 8B*). Likewise, the *sur2Δ yos9Δ* double mutant cells (28 generations) lived nearly as long as the *sur2Δ* (30 generations) and longer than the *yos9Δ* single mutant cells (*Figure 8B*). Thus, weakening the cER diffusion barrier

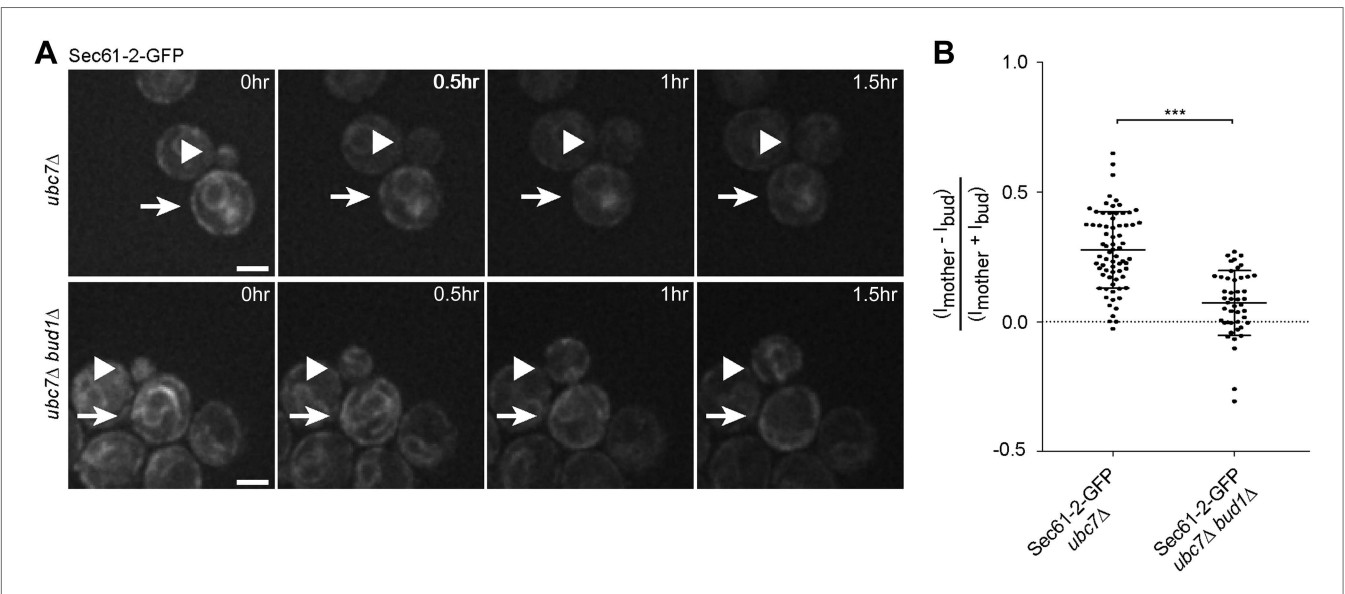

**Figure 7**. Misfolded Sec61-2-GFP asymmetry depends on the cER diffusion barrier. (**A**) Images of Sec61-2-GFP of a budding *ubc7Δ* (up) and a budding *ubc7Δ bud1Δ* (bottom) mutant cell. The arrow indicates the mother cell and the arrowhead the growing bud. Scale bar = 2 µm. (**B**) Distribution of asymmetry indices measured at the last frame prior to anaphase in individual cells of indicated genotype (note that the quantification of Sec61-2-GFP *ubc7Δ* is the same as in *Figure 1G*). ***p<0.0001 (*t* test).

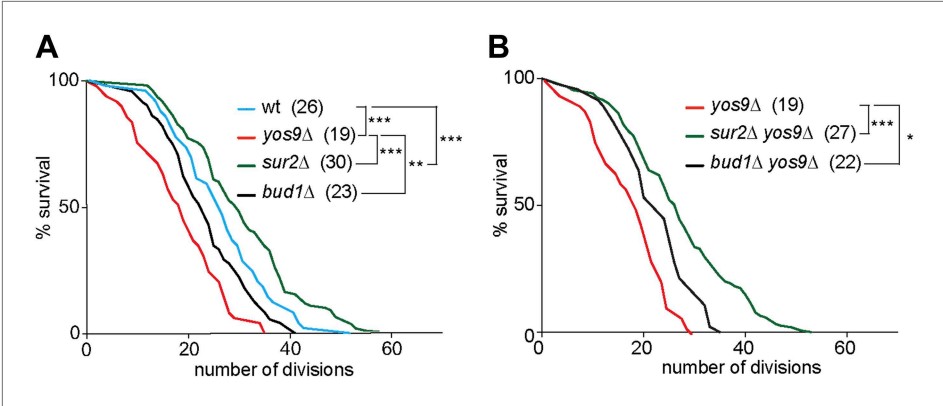

**Figure 8**. *Yos9* deletion shortens the life span of cells. (**A** and **B**) Pedigree analysis of wild type and mutant strains. Graph indicates the number of surviving cells after the indicated number of divisions. The median survival age for the tested strains is indicated n > 50 mother cells. p***<0.001, **p<0.01, *p<0.05 (Gehan-Breslow-Wilcoxon test).

is sufficient to suppress the toxic effect of the *YOS9* deletion on life span. However, we note that *sur2Δ* mutant cells are longer lived than wild type cells, whereas *bud1Δ* mutant cells are slightly shorter lived (*Figure 8A*). We previously showed that weakening the ER and the nuclear envelope diffusion barriers through the deletion of the *BUD6* gene extended life span, probably by passing on aging factors to the mother's cell progeny (*Shcheprova et al., 2008*). Here, Sur2 is required for the formation of both barriers, hence for the retention of aging factors from both the ER and the nucleus, whereas Bud1 is only required for the ER barrier. Since *sur2Δ* mutant cells are long-lived and *bud1Δ* mutant cell slightly short lived, it seems that the ER diffusion barrier is not required for the retention of the aging factors limiting the life span of cells growing under optimal conditions. The ER diffusion barrier seems to become critical for aging only upon induction of ER stress.

## Sphingolipids are downstream components of the diffusion barrier pathway

Together our data indicate that both the Bud1 module and sphingolipid biosynthesis contribute to the formation of the lateral diffusion barrier in the cortical ER, together with septins and Bud6 (*Luedeke et al., 2005*). In order to determine how these players function together, we first investigated whether they formed a single genetic pathway or defined several pathways by determining whether their inactivation had synergistic effects on barrier strength. Measuring the barrier indices of the *sur2Δ bud1Δ*, *sur2Δ bud6Δ* and *bud6Δ bud1Δ* double mutants indicated that all of them showed a similar defect and the BI was not lower than the lowest BI due to the individual mutations (BI = 2.6 ± 1.2 in *bud1Δ*, 3.9 ± 1.1 in *sur2Δ*, 3.8 ± 1.5 in *bud6Δ*, 3.3 ± 1.3 in *sur2Δ bud1Δ*, 3.5 ± 1.7 in *sur2Δ bud6Δ*, 3.5 ± 1.7 in *bud6Δ bud1Δ* single and double mutant cells, *Figure 9*). Thus, sphingolipids, Bud1 and Bud6 functioned in the same genetic pathway.

In order to position this pathway relative to septin function, we next investigated whether any of these proteins interfered with septin assembly. We characterized the localization of the septin Shs1, fused to GFP, in *sur2Δ*, *bud1Δ* and *bud6Δ* single mutant cells. In all these cells, the localization and intensity of Shs1-GFP at the bud neck was comparable to wild type (*Figure 10A*), suggesting that none of these factors acts upstream of septins.

Similarly, we asked whether the septins, Bud6 and sphingolipids functioned upstream of the Bud1 module. Since Bud1 localization is diffuse, the localization of its GEF, Bud5, is probably what determines where this GTPase is active. Therefore, we investigated whether any of the corresponding mutants affected the localization of Bud5. Bud5-GFP fluorescent intensity at the bud neck was clearly reduced in *shs1Δ* mutant cells but was not affected in *bud1Δ*, *sur2Δ* and *bud6Δ* single mutant cells (*Figure 10B*). To test whether Bud6 and Sur2 were acting redundantly on the localization of Bud5-GFP, we created the *sur2Δ bud6Δ BUD5:GFP* mutant strain. In these cells, the level of Bud5-GFP at the bud neck was comparable to the level observed in wild type cells (*Figure 10B*). The localization of Bud5-GFP was also intact in the *bud6Δ bud1Δ* and *sur2Δ bud1Δ* double mutant cells (*Figure 10B*). Thus,

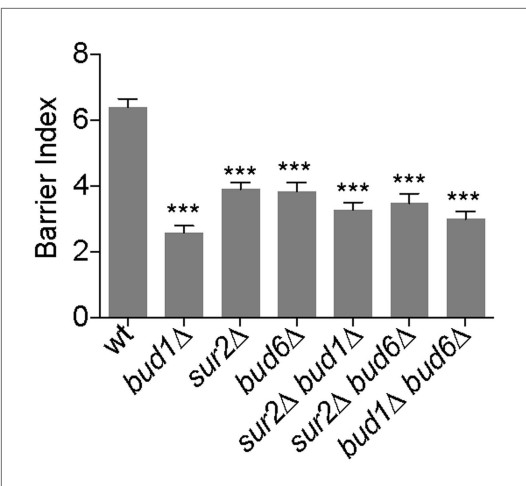

**Figure 9**. Bud1, Bud6 and Sur2 act in the same pathway for barrier formation. Graph shows the BI measured by FLIP analysis of Sec61-GFP exchange between mother and bud in cells of indicated genotype, as in *Figure 2*. BI + SD of tested strains are shown. n > 20 cells. ***p<0.001, (*t* test).

while septins control the localization of Bud5 and hence, the activation of Bud1 at the bud neck, Bud6 and Sur2 act downstream of Bud1.

Consistent with this interpretation, analysis of Bud6-GFP distribution indicated that proper Bud6 localization to the bud neck depends on Shs1, as reported (*Luedeke et al., 2005*), and on Bud1 function, but not on Sur2 (*Figure 10C*). Double deletion of *BUD1* and *SUR2* did not influence the fluorescence intensity of Bud6-GFP more than the deletion of *BUD1* alone, suggesting that Sur2 is downstream of Bud6 (*Figure 10C*). Taken together, these studies place septins at the top of a signaling pathway involved in the assembly of the diffusion barrier in the ER-membrane at the bud neck, and Bud6 and sphingolipids at the bottom. Bud1/Cdc24/Cdc42 are likely to target Bud6, which in turn regulates the sphingolipid-dependent formation of the diffusion barrier at the bud neck (*Figure 10D*; 'Discussion').

## ER membrane at the bud neck differs from the rest of the ER

Thus, Sur2 and hence the biosynthesis of sphingolipids, seemed to act most downstream in barrier formation. Furthermore, sphingolipid biosynthesis was among the only requisites conserved in both the cortical and the nuclear diffusion barriers. These observations suggested that the barrier might consist of a specialized lipid domain in the ER-membranes at the bud neck. We rationalized that the exclusion of the proteins Sec61 and Hmg1 from the bud neck region (*Luedeke et al., 2005*) might indicate that the ER-membrane composition is indeed distinct in this area. Therefore, we next wondered whether all ER membrane proteins were excluded from the ER-membrane at the bud neck. A set of such proteins, namely Lcb1, Sur2, Get1 and Ste24 was tagged with GFP in cells expressing the ER lumen marker dsRed-HDEL (*Onischenko et al., 2009*), and their localization to the bud neck was characterized. Lcb1 and Sur2 are both required for the biosynthesis of sphingolipids (*Dickson, 2008*), Get1 is involved in the insertion of tail anchored proteins into the ER membrane (*Schuldiner et al., 2008*) and Ste24 is a conserved zinc metallo-protease required for alpha-factor peptide maturation and CAAX-box protein processing (*Boyartchuk et al., 1997*; *Fujimura-Kamada et al., 1997*; *Tam et al., 1998*; *2001*). All of these proteins were excluded from the bud neck region, whereas dsRed-HDEL was not (*Figure 11A*). Therefore, all ER membrane proteins tested so far are excluded from the bud neck region. As previously shown (*Luedeke et al., 2005*), dsRed-HDEL localization showed that the ER is a continuous organelle between the mother and bud. These results suggest that the ER-membrane at the bud neck differs from the rest of the ER.

We next rationalized that sphingolipids might help barrier formation by forming this specialized membrane domain. To test this possibility, we asked whether disruption of sphingolipid biosynthesis affected the exclusion of trans-membrane proteins from the ER membrane at the bud neck. Therefore, we characterized the localization of Sec61-GFP at the bud neck in various barrier mutants. The phenotypes observed for individual cells were sorted in three categories (*Figure 11B*). In the group 1 (green) Sec61-GFP was totally excluded from the bud neck. At the other extreme, Sec61-GFP localization was continuous at the bud neck periphery in at least one side of the neck in the cells from the group 3 (red). The group 2 (grey) consisted of the cells with intermediate phenotypes. In wild type cells, 81% of cells totally excluded Sec61-GFP from the bud neck (*Figure 11B*). The *shs1Δ* mutant cells showed the most prominent loss of Sec61-GFP exclusion from the bud neck (~5%; *Figure 11C*). The *bud1Δ*, *bud5Δ*, *cdc24-4* and *lcb1-100* single mutant cells formed the second strongest set of mutants, showing a very strong defect in Sec61-GFP exclusion (between 27–35%, *Figure 11C*), closely followed by the *sur2Δ* and *bud6Δ* mutant cells (44–47% exclusion). In contrast, the lipid synthesis mutations *alg3Δ* and *gpt2Δ* did not affect Sec61-GFP localization much (69–70% exclusion, *Figure 11C*). Therefore, formation of

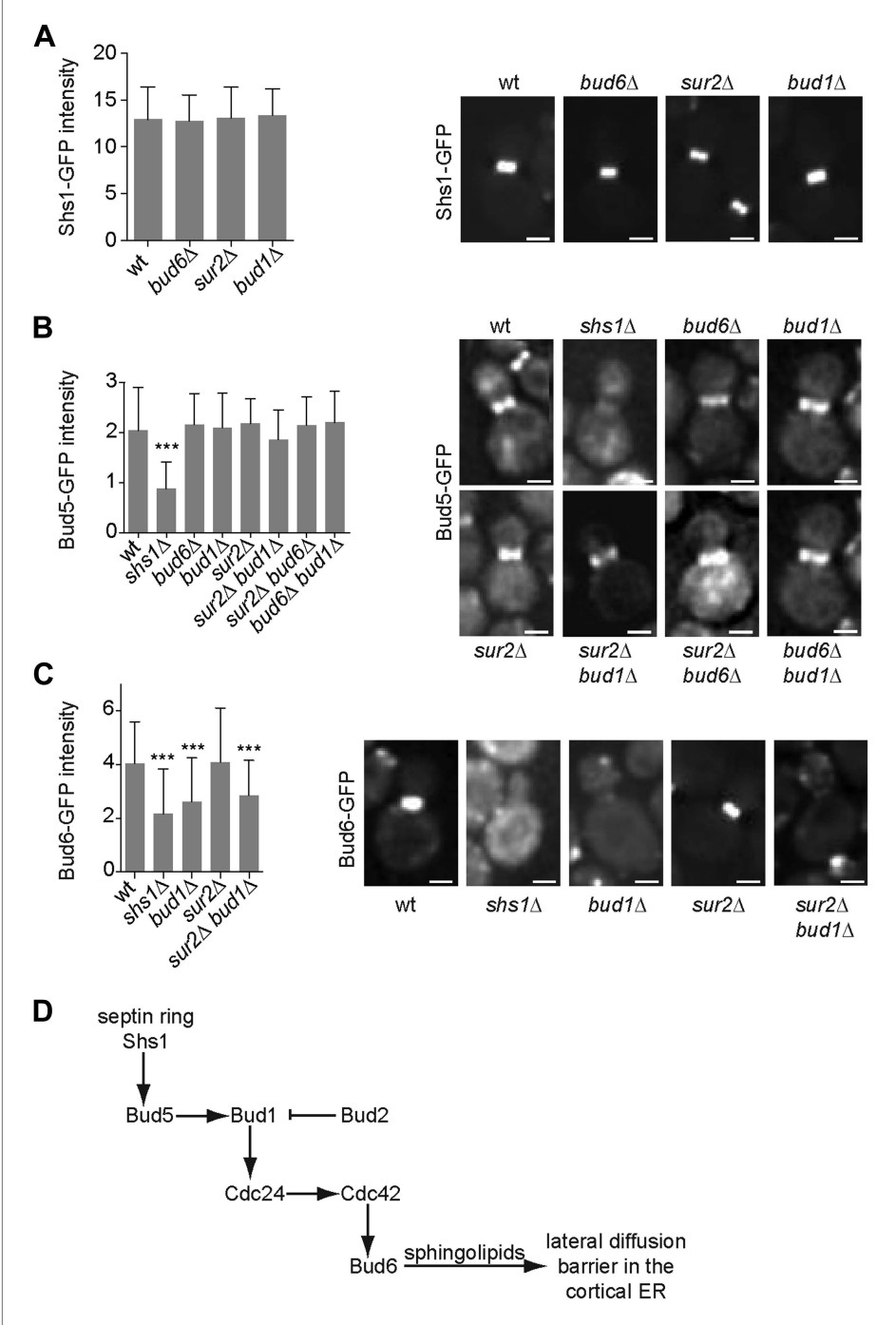

**Figure 10**. Epistasis analysis of the factors required for the assembly of the diffusion barrier in the cortical ER. (**A–C**) Fluorescence intensity of Shs1-GFP (**A**), Bud5-GFP (**B**) and Bud6-GFP (**C**) was measured at the bud neck by taking z-stacks of the cells after deconvolution with Softworx software. n > 100 cells. Representative images (right) and average fluorescence intensities are shown ± SD (arbitrary units, left). n > 20 cells. ***p<0.001, (*t* test). Scale bars = 2 µm. (**D**) Schematic drawing of the pathway required for the establishment of the diffusion barrier.

the exclusion zone in the ER-membrane at the bud neck is the most downstream event in the pathway of barrier formation, and the only event to depend on sphingolipid biosynthesis. We concluded that sphingolipids contribute to building a specialized ER membrane domain at the bud neck, in a Bud1-, Cdc42- and Bud6-dependent manner.

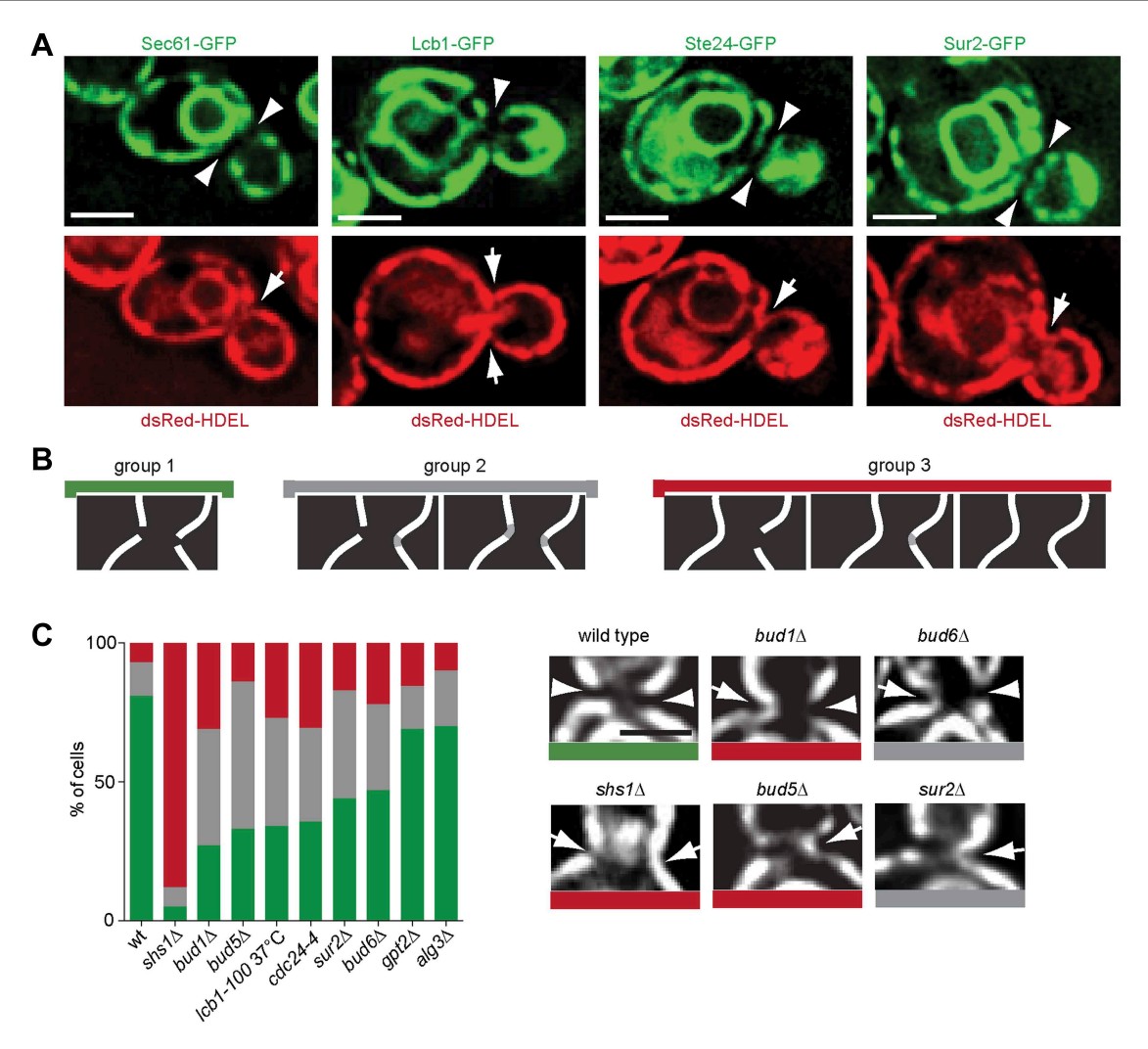

**Figure 11**. Exclusion of ER-transmembrane proteins at the cER bud neck. (**A**) localization of the indicated ER membrane resident proteins (green) and the ER luminal marker dsRed-HDEL (red) at the bud neck of the same cells. (**B**) Classification of Sec61-GFP exclusion from the bud neck in three groups: total exclusion (green), partial exclusion (grey) and no exclusion (red). (**C**) Evaluation of Sec61-GFP exclusion from the bud neck in wild type and mutant cells of indicated genotype, using the classification above. Arrowheads indicate exclusion, arrows indicate no exclusion. Representative cells are shown. n > 100 cells. Scale bars = 2 μm.

## Different lipid composition at the bud neck in the ER- and nuclear membrane

Taken together, our data suggested that the barrier may consist of a specialized lipid domain in the ER membrane at the bud neck, possibly involving sphingolipids. Thus, we next wondered whether we could observe a difference in lipid composition for the ER-membrane at the bud neck. Unfortunately, we were unable to identify a dye able to reliably visualize sphingolipids in yeast membranes. Instead we used the $DiOC_6$ (3,3'-dihexyloxacarbocyanine iodide) fluorescence dye, a cell-permeant, green-fluorescent dye that stains initially the ER membrane by binding to the hydrophilic head group of lipids before being transported to the mitochondria (*Terasaki, 1989*). Strikingly, $DiOC_6$ was excluded from the bud neck region of metaphase cells similarly to ER-membrane proteins (*Figure 12A*). Similar to Sec61-GFP, total exclusion of $DiOC_6$ from the bud neck was most strongly affected in the *shs1Δ* septin mutant cells (wt = 77%, *shs1Δ* = 10% exclusion), followed by the *lcb1-100*, and *bud6Δ* single mutant cells (38%, 40% exclusion, respectively, *Figure 12B*). The *sur2Δ, bud1Δ, bud5Δ* and *cdc24-4* mutations

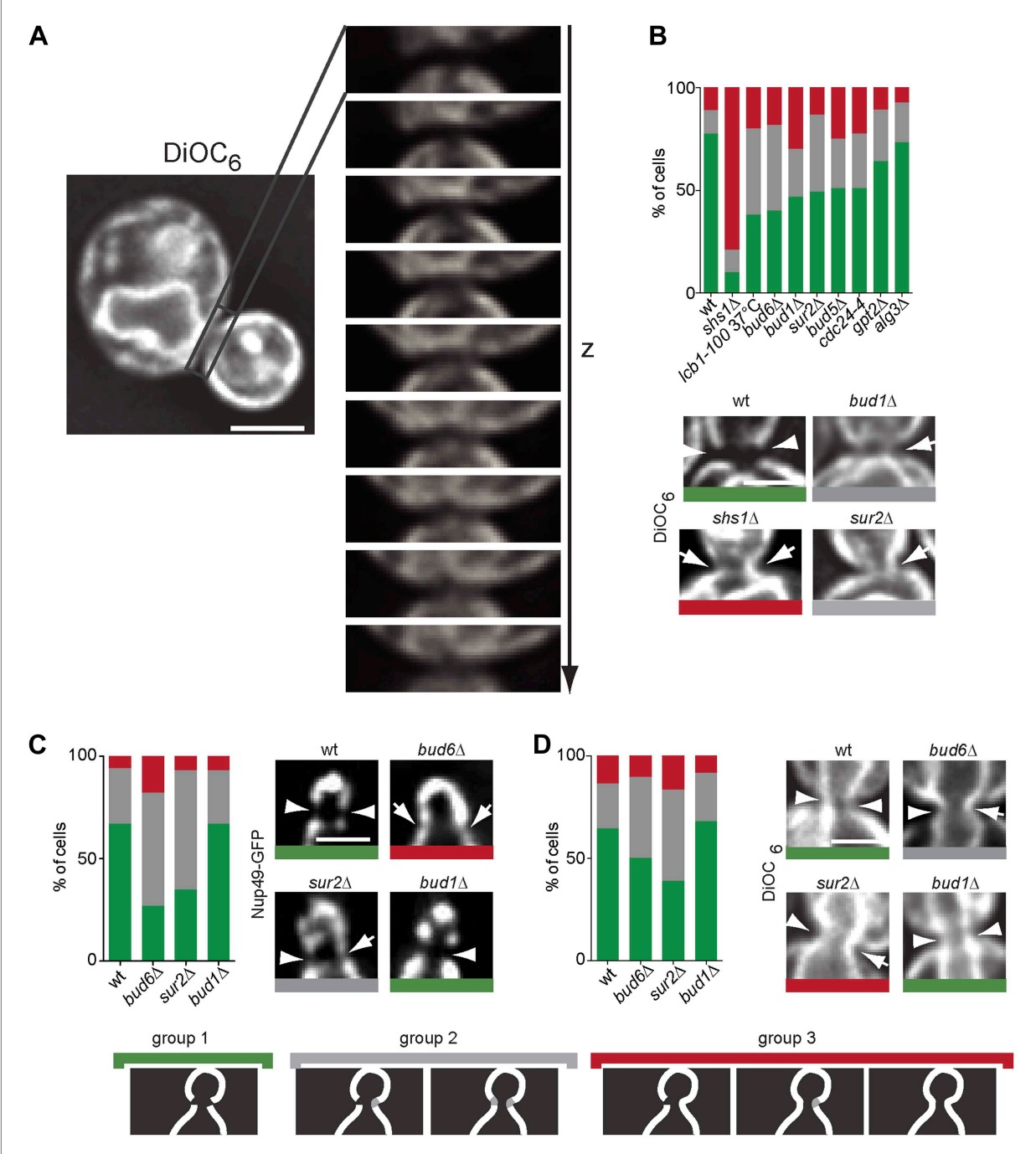

**Figure 12**. The composition of the cER and Nuclear membrane at the bud neck differs from elsewhere. (**A**) A representative metaphase cell stained with the dye DiOC$_6$ and images at different z-positions through its bud neck are shown. (**B**) Evaluation of DiOC$_6$ exclusion at the bud neck in wild type and mutant cells according to the classification used for Sec61-GFP in *Figure 11B*. (**C–D**) Characterization of the exclusion of Nup49-GFP (**C**) and DiOC$_6$ (**D**) from the bud neck in early anaphase nuclei of cells of indicated phenotype using the same classification principle as in *Figure 11B*. Representative cells illustrated. n > 100 cells. Scale bars = 2 μm.

caused a slightly weaker phenotype (between 46–50% exclusion), whereas the *alg3Δ* and *gpt2Δ* mutations showed the weakest or no phenotype (between 64–73% exclusion, *Figure 12B*). Thus, the entire barrier pathway was required for the exclusion of the lipid dye DiOC6 from the ER membrane at the bud neck, including sphingolipids, supporting the conclusion that barrier formation involves the

assembly of a specialized membrane domain of unique lipid composition in the ER membrane at the bud neck.

Since our data established that formation of the barrier in the nuclear envelope also depends on sphingolipid biosynthesis, we wondered whether a specialized lipid domain of the outer nuclear membrane formed this second barrier as well. Analysis of nuclear pore distribution, using Nup49-GFP as a reporter, indicated that in wild type, anaphase cells the abundance of NPCs was indeed reduced in the nuclear envelope at its intersection with the plane of the bud neck (67% exclusion, *Figure 12C*). Exclusion depended on Bud6 and Sur2 (27% and 35% exclusion, respectively), but not on Bud1 (67% exclusion, *Figure 10C*). Similarly, a fainter, but significant effect was also observed for DiOC6 staining in wild type, anaphase cells (*Figure 12D*). Reduced DiOC6 staining in the nuclear envelope at the bud neck (64% of wild type cells) partially depended on Bud6 and Sur2 function (38 and 50% respectively), but not on Bud1 (68%, *Figure 12D*). These observations are consistent with Bud6 and sphingolipids being required for the formation of a specialized membrane domain at the location of the barrier in the nuclear envelope. They are also fully consistent with our data establishing that Bud1 is not involved in the assembly of the diffusion barrier in the outer nuclear membrane.

## Discussion

Although recent studies have underlined their broad distribution and roles, we still know little, if anything, about how lateral diffusion barriers form in eukaryotes, what they are composed of and how they function. To start addressing these questions, we investigated the organization and function of ER barriers in budding yeast. This study establishes several points. Firstly, they reveal the importance of sphingolipids in the organization and compartmentalization of the ER. Together with the observation that most membrane proteins are partially excluded from the barrier area, our data suggest a mechanism for how barriers form and function, (*Figure 13* and 'Diffusion barriers as specialized lipid domains of the ER membrane'). Secondly, our study indicates that different barriers are probably based on similar principles, yet their formation is dictated by different signaling mechanisms. We have identified some of the signaling processes involved in the assembly of the diffusion barrier in the cortical ER, namely the Bud1 and the Cdc24 module. However, the nuclear envelope diffusion barrier is not affected in the *bud1Δ* mutant cells, indicating that assembly of the nuclear barrier is controlled differently. Thirdly, this difference in regulation allows the barriers to be studied individually to assess their specific functions. In general, these studies underline the importance of lateral diffusion barriers in the confinement of cellular stress to the mother cell. They also allow distinguishing the relative impact of different types of stresses on cellular aging under different growth conditions. We discuss below each of these three aspects of our findings in more detail.

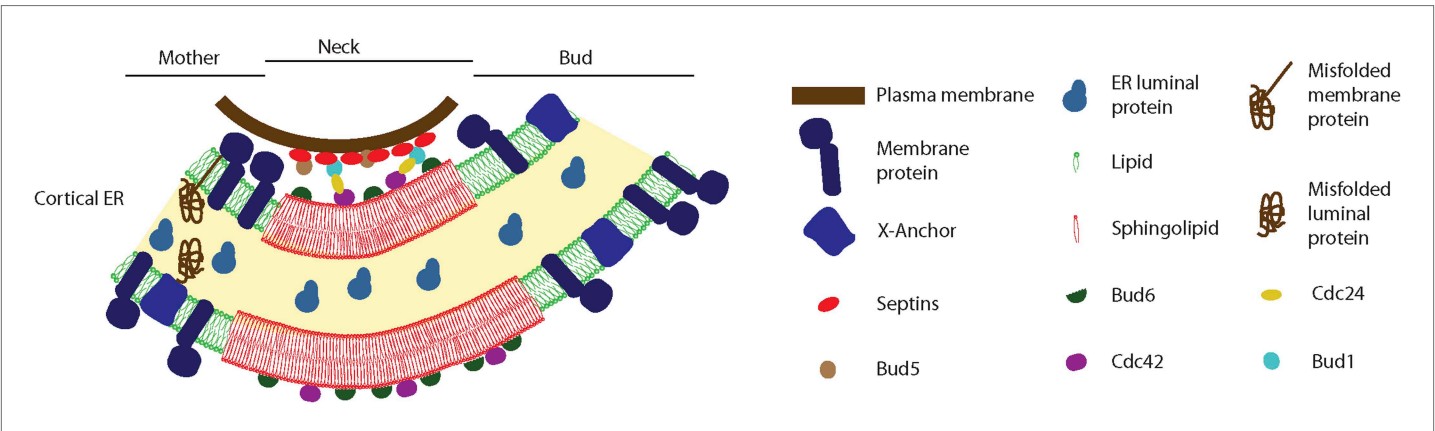

**Figure 13**. Model for the cortical ER diffusion barrier. Septins (red ovals) assemble at the plasma membrane and organize the Bud1-Cdc42 module (brown, blue, yellow and violet ovals), which targets Bud6 (green). Bud6 might help organizing sphingolipids (red) at the bud neck in a liquid-ordered domain, thereby forming a lateral diffusion barrier. Note that trans-membrane proteins (dark blue) are excluded from this region but not from the rest or the ER (green lipids), due to their short transmembrane domain. ER luminal proteins (light blue) are found in the whole ER lumen. Misfolded proteins are retained in the mother part of the ER because they are themselves trans-membrane proteins or because they are attached to an unknown anchor (X, blue).

## Different roles in age retention for different barriers

While mother cells age with each division, the buds are born with a full life-span expectancy. Replicative aging of the mother is thought to be caused by the accumulation of damages, which they do not share with their daughters. However, what these different factors are and how they are retained in the mother cell is unclear. Here, we show that misfolded ER-proteins are retained in the mother cell by two distinct mechanisms: the formation of nearly immobile aggregates, and the retention of smaller entities by the diffusion barrier in the cortical ER membrane at the bud neck. Interestingly, the fact that the barrier in the ER membrane helps retaining individual, misfolded proteins in the mother cell, although many of them, including CPY*, are luminal, indicates that an unidentified receptor may anchor them in the membrane. Under optimal growth conditions, wild type cells appear to contain only little misfolded proteins and ER stress does not appear to be the main cause of their demise with age. Indeed, if ER stress were a constitutive cause of aging, disruption of the barrier in the cortical ER should systematically increase the longevity of the cells, which is not the case. However, low concentrations of tunicamycin or affecting protein quality control increased the load of misfolded proteins, and accelerated aging. Allowing mothers to pass stress on to their daughters by down regulating the ER diffusion barrier diluted this stress, as shown by the fact that it reduced the load in the mother cells and restored the life span of the *yos9Δ* mother cells. Thus, retention of ER-misfolded proteins in the mother cell caused stress and accelerated aging. ER stress may therefore be considered as a conditional aging factor. This might be particularly the case in highly secreting cells, such as beta-pancreatic cells in mammals (*Kennedy et al., 2010*).

Interestingly, affecting all barriers by disrupting the general barrier assembly factors Bud6 and Sur2 extends the longevity of the cell even under optimal growth conditions. Since this effect is not due to loss of the barrier in the cortical ER, we assume that it could be due to the loss of the nuclear barrier, as previously suggested (*Shcheprova et al., 2008*).

## Differential regulation of distinct diffusion barriers

In this study we confirm that the septin ring at the bud neck lies upstream in the barrier formation pathway, at least in the cortical ER. But it was unclear how septins, which are closely apposed to the plasma membrane, promote formation of a barrier on both sides of the ER lumen. Two observations may help solving this conundrum. First, the role of septins in barrier assembly is partly indirect. The fact that the GTPases Bud1 and Cdc42 contribute to barrier formation downstream of septins suggests that the localization signal originating at the plasma membrane spreads from there to specialize the ER-membrane at the bud neck (*Figure 13*). This signal regulates the assembly of a specialized membrane domain, which itself functions as a barrier. However, none of the mutants characterized shows as strong a phenotype as the septin mutant *shs1Δ*. Therefore, septins might function above more than only one pathway in barrier assembly. Future studies will be needed to address this possibility.

Together, our findings suggest that the pathway for barrier formation consists of three successive layers. At the top, the septins provide a spatial cue for where the barrier should form, functioning as a scaffold for regulatory elements.

Downstream of septins, a second layer transduces the localization signal to effectors. This module depends at least in part on GTPases, namely Bud1 and Cdc42 in the case of the cortical ER barrier. Unlike during bud-site selection, Bud1's role in barrier formation might not be restricted to the early times of bud emergence. Indeed, Bud1's activator, Bud5, remains at the bud neck throughout bud growth. Thus, barrier assembly might be a continuous process, also ensuring barrier maintenance.

The second layer is specific to the barrier under consideration. Whereas the Bud1/Cdc42 module controls the cortical barrier, we do not know at this time which signaling molecules govern the formation of the barrier in the nuclear envelope.

Downstream of this second layer, a third component of the pathway is more directly involved in barrier formation. This layer appears to be conserved between the different barriers in the ER, supporting the idea that it plays a structural role. This is where we find the protein Bud6, a target of Cdc42 (*Jaquenoud and Peter, 2000*), and the sphingolipids. Future characterization of this layer will provide biophysical understanding about how barriers function.

## Diffusion barriers as specialized lipid domains of the ER membrane

Recent electron tomography studies characterized the yeast ER in detail and confirmed that ER is present in the bud-neck and that the yeast ER is physically continuous throughout the cell (*West et al.,*

*2011*). However, using morphological criteria, these studies failed to observe ER at the cortex of the bud neck. Although the luminal marker dsRed-HDEL confirms the presence of the ER at the cortex of the bud neck, Sec61 (*Luedeke et al., 2005*) and all the ER-membrane protein that we visualized, as well as the lipid dye DiOC6 are excluded from the ER in this area. The poor staining of the ER membrane at the bud neck by DiOC6, relative to the rest of the ER, suggests that the lipid composition of the membrane differs at that position. This could explain why the ER-membrane at the cortex of the bud neck was not identified in the tomography data.

The fact that sphingolipids play such an important, specific role at the bottom of the pathway for the formation of all ER barriers makes them prime candidates for structural elements of the barrier. Complex sphingolipids are thought to reside primarily in the plasma membrane, yet many of the enzymes involved at least in the first steps of their biogenesis localize to the ER, including Lcb1 and Sur2. Thus, it would be in principle possible that an ER-specific pool of sphingolipids is directly involved in barrier formation. If this were the case, it would explain why DiOC6 is excluded from the ER-membrane in the barrier areas. DiOC6 stains membranes essentially through interaction with hydrophilic heads of lipids, and shows little affinity for sphingolipid-rich membranes. Thus, DiOC6 exclusion from the ER membrane at the bud neck would be compatible with sphingolipid accumulation at this location.

Sphingolipids are characterized by long side chains. The idea that such lipids form a core component of lateral diffusion barriers is attractive for several reasons. The long fatty acid chains would promote the formation of a thicker lipid bilayer. This would explain why proteins with transmembrane domains tailored to fit the average thickness of the ER membrane are excluded from the barrier area. Furthermore, long-chain sphingolipids are thought to be prone to form liquid ordered domains (*Aresta-Branco et al., 2011*), promoting their separation from other lipids. In this area, the membrane would be predicted to be less fluid and to disperse only slowly, two ideal conditions for the formation of a localized barrier. However, it is very unlikely that this domain would be stable by itself. Therefore, it is likely that the barrier must be continuously reassembled and thus would comprise protein components to stabilize it. Bud6 is an excellent candidate for such a function. Indeed, its localization to the bud neck does not depend on sphingolipids, yet Bud6, which is a peripheral membrane protein (*Jin and Amberg, 2000*), acts in forming the specialized membrane domain. Thus, one of Bud6 function may be to stabilize a sphingolipid domain in the ER membrane at the bud neck, probably together with other not yet identified proteins. Actin would be here an excellent candidate, given the role of Bud6 in actin cable formation and actin binding. However attractive this model is, we have so far no dye that stains sphingolipids in yeast, such that we are not in the position yet to determine whether sphingolipids indeed accumulate in the ER-membrane at the bud neck. Identifying sphingolipid-binding proteins will probably be necessary in order to solve this issue. Future studies will also need to identify the exact identity of the sphingolipids involved.

Taken together, our data provide molecular insights into how lateral diffusion barrier might assemble and function in eukaryotic membranes. These insights set the stage for investigating how barriers function at the biophysical level, and whether their mechanism is conserved in other organisms. Indeed, it is remarkable that diffusion barriers of the plasma membrane are highly conserved, on a molecular level as demonstrated by the involvement of septins in many of them. However, we still know little about whether the ER is also compartmentalized in other eukaryotes. Based on the function of ER barriers in yeast, namely the asymmetric segregation of damages and aging determinants, it is tempting to assume that similar barriers will be found to play similar roles in a wide variety of other cell types, and particularly in cells that show strong self renewal.

## Material and methods

### Strains, plasmids and growth conditions

All yeast strains were constructed according to standard genetic techniques and are isogenic to S288C, unless stated otherwise. The different deletions were obtained from the EUROSCARF deletion collection (Frankfurt, Germany) and provided to us by M Peter (ETH, Zurich, Switzerland). All experiments were performed with three independent clones. All cultures or plates were grown at 30°C.

### FLIP and FRAP experiments

For all FLIP and FRAP experiments, fresh cells were grown at 30°C on YPD (yeast, peptone, and 2% dextrose) plates, resuspended in synthetic complete medium, and immobilized on a 2% agar pad

containing synthetic complete medium. The cells were imaged on a confocal microscope (LSM 510; Carl Zeiss, Jena, Germany) with a Plan Apochromat 63 × /1.4 NA oil immersion objective, using 2% of argon laser intensity (488 nm line) at 40% output. The ZEN 2010 software (Carl Zeiss) was used to control the microscope. GFP emission was detected with a 505 nm long pass filter. Photobleaching was applied on a region of interest (ROI) as indicated in the figures. For FLIP experiments, the bleaching regions were irradiated with 80 iterations at 80% laser power and 40% transmission over a period of 40 frames. For FRAP experiments, bleaching was applied once with 60 iterations in the bud compartment or 70 iterations in the mother compartment at 80% laser power and 40% transmission. All photobleaching experiments were performed at 30°C, unless stated otherwise.

FLIP quantification were performed using ImageJ 1.42q (National Institutes of Health). The mean fluorescence signal was quantified in the entire mother cell, the entire bud and in the entire mother of five neighboring cells. After background subtraction, the fluorescence signals of the mother and bud were normalized to the mean of the five control cells and set to 100% at beginning of the experiment. All experiments were pooled and transferred to Prism 5.0b (GraphPad Software, La Jolla, California), in which a one-phase decay curve constraining the first bleaching point to 100% was fitted. The Barrier Index (BI) was defined as the ratio of the times needed to lose 50% of the fluorescence signal in the bud over the mother. All images shown in the figures were processed using ImageJ 1.42q.

FRAP quantification were also performed using ImageJ 1.42q. The mean fluorescence recovery signal was quantified in the bleached ROI in either the mother or bud compartment and as a control the same size region was measured in five neighboring cells. After background subtraction, the fluorescence signals of the mother and bud were normalized to the mean of the five control cells and set to 100% at beginning of the experiment. All experiments were pooled and transferred to Prism 5.0b and on an exponential FRAP curve, the mobile fraction was measured by determining the half time ($t_{50}$) of fluorescence recovery to reach a plateau level.

### Exclusion of Sec61-GFP, dsRed-HDEL, DiOC6 and Nup49-GFP

To determine the exclusion of Sec61-GFP, dsRed-HDEL, DiOC6 and Nup49-GFP, we took 30 z-stacks (0.2 µm steps) on a Deltavision microscope (Applied Precision, GE Healthcare company, Issaquah, Washington) equipped with a CoolSNAP HQ$^2$ camera (Photometrics) and an Olympus (Tokyo, Japan) 100 × oil immersion objective and the GFP/Cherry filter set. A GC400 filter was used to eliminate UV. The microscope was controlled by the SoftWorks software. All quantifications were performed using ImageJ software.

### Fluorescence intensity levels measurements of bud neck proteins

To determine the fluorescence intensity of bud neck protein for the epistasis experiments, we took 30 z-stacks (0.2 µm steps) on the Deltavision microscope (Applied Precision) equipped with a CoolSNAP HQ$^2$ camera (Photometrics, Tucson, Arizona) and an Olympus 100 × oil immersion objective and the GFP/Cherry filter set. A GC400 filter was used to eliminate UV. The microscope was controlled by the SoftWorks software. All quantifications were performed using ImageJ software.

### Visualization of GAL-CPY*-GFP overexpression

Cells containing the *GAL-CPY*-GFP* gene were grown at 30°C in synthetic media containing the appropriate amino acids and 2% galactose for 6 hr. To visualize the expression of CPY*-GFP, we took 30 z-stacks (0.2 µm steps) on the Deltavision microscope (Applied Precision) equipped with a CoolSNAP HQ$^2$ camera (Photometrics) and an Olympus 100 × oil immersion objective and the GFP/Cherry filter set. A GC400 filter was used to eliminate UV. The microscope was controlled by the SoftWorks software. All quantifications were performed using ImageJ software. Time-lapse microscopy was performed in a Y04C microfluidic chamber (Cellasic, Hayward, California) controlled by an ONIX microfluidic perfusion platform. Synthetic media was flowed in the chamber at 1 Psi.

### Photoconversion experiments

Sec61 was endogenously tagged with tdEOS as described (*McKinney et al., 2009*). Cells were imaged using a laser scanning confocal microscope LSM 780 (Zeiss) equipped with a high sensitive multiarray 32PMT GaAsP detector. Photoconversion was applied on a ROI as indicated in the figures (half of the mother cell). For photoconversion a 405 nm laser at 10% laser intensity was used. The ROI was converted once with 20 iterations. After conversion an image was taken every 20 s for 2 min. Images were analyzed using ImageJ. Integrated fluorescence intensity of the converted fluorophore (red) was

measured in the bud for every time point and after subtraction of the background normalized to 10 for the first image (before conversion). An equally normalized unconverted control cell was then subtracted. The average normalized fluorescence intensity of all cells and the standard error of the mean was plotted in the graph.

## Life span analysis

Lifespan analyses were carried out as previously described (*Kennedy et al., 1994*).

## Acknowledgements

We would like to thank Karsten Weis (UC Berkeley), Matthias Peter (ETH, Zürich), Robert Gauss, Markus Aebi (ETH, Zürich) and Howard Riezman (University of Geneva) for sharing strains, the Scientific Center for Optical and Electron Microscopy ScopeM of the Swiss Federal Institute of Technology ETHZ for microscopy support and Markus Aebi, Andreas Conzelmann, Roger Schneiter, Claudio De Virgilio, Christine Weirich and Juha Saarikangas for discussions and suggestions.

## Additional information

### Funding

| Funder | Grant reference number | Author |
|---|---|---|
| Swiss National Science Foundation | Synergia CRSI33_125232/1 | Lori Clay, Yves Barral |
| European Research Council | BarrAge 250278 | Lori Clay, Fabrice Caudron, Annina Denoth-Lippuner, Barbara Boettcher, Yves Barral |
| Swiss Federal Institute of Technology Zürich | TH-33 06-1 | Barbara Boettcher |

The funders had no role in study design, data collection and interpretation, or the decision to submit the work for publication.

### Author contributions

LC, FC, Acquisition, analysis and interpretation of data, Writing of the article; AD-L, Acquisition, analysis and interpretation of data, Critical reading of the draft; BB, Acquisition and analysis of data; SBF, Critical reading of the manuscript, Analysis and interpretation of data; ELS, Contributed reagents, Interpretation of data, Critical reading of the draft; YB, Conception and design, Interpretation of data, Writing of the article

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
