## [Decision Letter]

Thank you for sending your work entitled “A sphingolipid-dependent diffusion barrier confines ER stress to the yeast mother cell” for consideration at *eLife*. Your article has been favorably evaluated by a Senior editor and 3 reviewers, one of whom is a member of our Board of Reviewing Editors.

The Reviewing editor and the other reviewers discussed their comments before we reached this decision, and the Reviewing editor has assembled the following comments to help you prepare a revised submission.

Previous work by Barral and colleagues has shown that a lateral diffusion barrier at the neck region of the budding yeast (*S. cerevisiae*) restricts the mobility of ER membrane proteins (Luedeke et al, J Cell Biol 169:897, 2005) and the passage of nuclear membrane-bound DNA circles from the mother cell to the bud, which leads to an age asymmetry between mother and bud (Shcheprova et al, Nature 454:728, 2008). The nature of the diffusion barrier and the signaling pathways and factors that mediate its formation are poorly understood. This study sheds light on the formation and lipid composition of the nuclear and ER membrane diffusion barrier, and shows that misfolded and presumably aggregated proteins are also asymmetrically retained within the mother cell. The main findings regarding the hierarchical roles of septins, Bud/Cdc42 signaling and sphingolipid biosynthesis for the formation of the diffusion barrier are well substantiated and represent a significant advance in the field.

Overall, the reviewers agreed that the combination of genetics and live cell analysis of protein dynamics in cells provides evidence of a sphingolipid-based diffusion barrier that is formed by proteins in the retention of candidate mis-folded proteins in the mother cell. Nevertheless, they also agree that you should consider the criticisms and suggestions discussed below which may help to strengthen the work further. In general all of the major criticisms are worth addressing, but you should pay special attention to the questions in #3.

Major criticisms and suggestions:

1) Given the emphasis on the asymmetric retention of ER misfolded proteins, the authors may consider assaying for diffusion and compartmentalization of more than one misfolded protein. The conclusions are based solely on FRAP studies of one chaperone (BiP) under conditions of ER stress and the localization of one misfolded protein (CPY*), whose diffusion was not investigated by FRAP. If Kar2 (BiP) is freely diffusible between the mother and bud compartments and there is no asymmetric accumulation of Kar2 within the mother compartment, it is unclear why misfolded proteins (e.g., CPY*, TM treatment) are retained within the mother cell and not the bud. No FLIP experiments are performed to address a potential asymmetry in Kar2 distribution within the mother compartment.

2) It's unclear how the authors envisage the selective accumulation of Kar2+misfolded proteins within the mother compartment. Do mis-folded proteins diffuse between the two compartments until they become big enough (i.e., large aggregates) within the mother compartment that they cannot diffuse back out to the bud? Can this be tested by assaying the movement of fluorescence molecules of variable size by selectively microinjecting or photo-activating them in mother/bud compartments? Overall, it is important to clarify the nature of the misfolded protein diffusion limitation. Are these protein aggregates or are they somehow crosslinking Kar2 into large complexes, or perhaps tethering Kar2 complexes to the lumenal surface of the ER. The use of Kar2 is elegant but indirect and they could readily corroborate this by imaging one of several model aggregation-prone proteins.

3) The authors do not provide any data on the percentage of recovery, which is an indication of the mobile fraction of Kar2 proteins within the ER. From the FRAP curves, it seems that there is more than a 50% reduction in the mobile fraction of Kar2-GFP after TM treatment within the mother cell. Does this reflect the formation of big aggregates that are virtually immobile or tethering to immobile elements of the ER?

4) Asymmetric retention of mis-folded proteins within the mother compartment is likely to be a time-dependent phenomenon. Perhaps the authors can gain a better insight by performing time-lapse experiments of CPY*-GFP movement between the mother and bud compartments and eventual retention within the mother? Does it seem that accumulation within the mother correlates with the gradual appearance of larger CPY* puncta within the bud? It might be also informative to FRAP CPY*-GFP and/or perform the FRAP of Kar2-GFP at various time-points during TM treatment.

5) The authors address the role of mis-folded protein retention in the longevity of the mother cell. They report that *sur2Δ* and *bud1Δ* mutants had longer and shorter life spans, respectively (Figure 6). They conclude that “…the cortical ER diffusion barrier does not participate in the retention of aging factors, …but that the nuclear barrier does.” They then report that a mutant in the ER quality control lectin Yos9 (*yos9Δ*) also resulted in shorter longevity. Significantly, *sur2Δ* and *bud1Δ* mutants in a *yos9Δ* background had ∼normal life-spans, resulting in the authors concluding that retention of mis-folded proteins in the ER of the mother cell accelerate its aging and hence identifies the importance of the cortical ER diffusion barrier. These two sets of experiments appear, especially as written, to be contradictory. Perhaps some re-writing would be helpful.

---

## [Author Response]

We have now addressed all the reviewer comments and added corresponding data to a revised manuscript. We feel that this new version has substantially gained in clarity and that our conclusions are now even better substantiated. Particularly, we have now added data regarding the compartmentalization of another misfolded protein, the temperature sensitive mutant Sec61-2. The distribution of this protein is also biased towards the mother cell, in a diffusion barrier dependent manner. Furthermore, we now provide FRAP data and time-lapse movies for CPY*-GFP. We believe that these additions extensively clarify its behavior and the mechanisms of its accumulation in the mother cell. The text was edited to become more detailed where needed, and simpler otherwise.

*1) Given the emphasis on the asymmetric retention of ER misfolded proteins, the authors may consider assaying for diffusion and compartmentalization of more than one misfolded protein. The conclusions are based solely on FRAP studies of one chaperone (BiP) under conditions of ER stress and the localization of one misfolded protein (CPY*), whose diffusion was not investigated by FRAP. If Kar2 (BiP) is freely diffusible between the mother and bud compartments and there is no asymmetric accumulation of Kar2 within the mother compartment, it is unclear why misfolded proteins (e.g., CPY*, TM treatment) are retained within the mother cell and not the bud. No FLIP experiments are performed to address a potential asymmetry in Kar2 distribution within the mother compartment*.

Following the reviewers' suggestion, we assayed the compartmentalization of another misfolded protein, a point mutant of the translocon protein Sec61. The protein encoded by the sec61-2 allele is known to misfold and to be sent for degradation by ERAD at restrictive temperature (Bordallo, Plemper, Finger, & Wolf, 1998). Already at permissive temperature, the levels of Sec61-2-GFP were much lower than those of wild type Sec61-GFP, suggesting that misfolding and degradation already happened. In an ERAD mutant (here *ubc7Δ*), misfolded Sec61-2 (8) and Sec61-2-GFP (this study) are degraded much slower, allowing to look at its distribution. Measuring the cortical ER fluorescence intensity, we observed that Sec61-2-GFP is retained is the mother part (asymmetry index of 0.27 in *ubc7Δ* and 0.12 in wild-type), whereas wild type Sec61-GFP is symmetrically distributed (asymmetry index of 0.00). Therefore, when misfolded this ER-transmembrane protein is also retained in the mother cell during asymmetric cell division. Note that in this case we do not observe the formation of aggregates as is the case for CPY*-GFP. This suggests that misfolded proteins do not require their aggregation or at least the formation of large structures to be selectively retained in the mother cell.

Concerning the diffusion of CPY*-GFP, FRAP assays (shown in Figure 1) show that the aggregates do not diffuse and their content only very slowly exchanges with the soluble fraction. We conclude from these observations that the barrier does not primarily retain the aggregates themselves (they stay put where they are formed), but the misfolded proteins prior to aggregation. Hence, aggregation takes place in the retention compartment.

Please, note that all experiments were done under mild stress conditions, where the cells continued budding at a relatively normal speed. If instead we acutely stress the cells, for example using high tunicamycin concentrations, misfolding takes place in both the mother and the bud and BiP is very rapidly slowed down in both compartments. Thus, the slowing-down of BiP and misfolded CPY*-GFP observed in the mother cell are most likely not due to retro-transport of stress from the bud into the mother cell, but more probably to the fact that the stress confined into the mother accumulates there over the time while the buds grow protected from this stress.

In favor of this interpretation, we provide movies of cells budding while expressing CPY*-GFP and Sec61-2-GFP (Figures 6 and 7, novel). As can be observed, while the bud grows, the CPY* aggregates already existing remain in the mother cell and increase in intensity, whereas the bud forms its own aggregates only later. For what concerns Sec61-2-GFP, at permissive temperature the cells grow buds that inherit much less of the protein than what is present in their mother. Again, equilibration takes place, but after division is completed. In both cases, elimination of the diffusion barrier using the *bud1Δ* mutation speeds up equilibration, which takes place now during bud growth and prior to the separation of the bud from its mother.

Finally, concerning the diffusion of BiP, we have characterized its exchange between the mother and the bud. We observe that already in the absence of stress BiP does not exchange particularly fast between the two compartments, consistent with the idea that it might be associated with a membrane protein, possibly Ire1, or folding substrates associated with the translocon or other receptors (our protein X). This data is not adding much to the FRAP and therefore has not been added to the manuscript.

In summary, we do not think that the accumulation of misfolded protein in the mother is due to their targeted aggregation in the mother or their retrotransport to the mother but more to the fact that the barrier limits their leakage from the mother into the bud. Thus, we suggest that the mother has more of it simply because it is older and because it remains the main site of synthesis during a large fraction of bud growth.

*2) It's unclear how the authors envisage the selective accumulation of Kar2+misfolded proteins within the mother compartment. Do mis-folded proteins diffuse between the two compartments until they become big enough (i.e., large aggregates) within the mother compartment that they cannot diffuse back out to the bud? Can this be tested by assaying the movement of fluorescence molecules of variable size by selectively microinjecting or photo-activating them in mother/bud compartments? Overall, it is important to clarify the nature of the misfolded protein diffusion limitation. Are these protein aggregates or are they somehow crosslinking Kar2 into large complexes, or perhaps tethering Kar2 complexes to the lumenal surface of the ER. The use of Kar2 is elegant but indirect and they could readily corroborate this by imaging one of several model aggregation-prone proteins*.

As summarized above, we do not see any difference between mother and bud immediately after stress or upon high stress, but only when cells are allowed to bud under mild stress conditions. Testing this idea, we now added data of cells exposed to tunicamycin for two different amount of time. Supporting our view, we observe that in the mother compartment the dynamics of Kar2 are slower and slower as the duration of tunicamycin treatment increases, while it remains fast in the buds emerging from these mothers (Figure 1). Thus, these data are supporting the idea that the asymmetry between the mother and the bud builds with time due to the retention of material in the mother.

We assume that aggregates form at a slow rate in both mother and bud but cannot exchange between these two compartments. Furthermore, since the aggregates themselves barely move, we assume that the barrier retains already small entities, such as perhaps the individual mis-folded proteins, as discussed above.

*3) The authors do not provide any data on the percentage of recovery, which is an indication of the mobile fraction of Kar2 proteins within the ER. From the FRAP curves, it seems that there is more than a 50% reduction in the mobile fraction of Kar2-GFP after TM treatment within the mother cell. Does this reflect the formation of big aggregates that are virtually immobile or tethering to*
*immobile elements of the ER?*

We have now looked into this more carefully. It is difficult to give precise information because in the long-term the proportion of recovery as well as recovery speed are limited by the exchange between mother and bud, the amount of fluorescence bleached in the first place, and the fraction trapped in the immobile fraction. Thus, all what we can say is that indeed the recoverable fraction drops in stressed cells, indicating that indeed a substantial fraction of Kar2 is immobilized in response to stress. We added a two-phase association analysis of Kar2-sfGFP FRAP curves of untreated mother cells and mother cells treated with Tm. We observe that in the untreated cells the fraction of fast moving particles is rather high (65.7%) while it becomes low in cells treated with tunicamycin (19.0%). About whether this reflects immobilization into big aggregates or tethering to other structures is difficult to tell. We observe that there are indeed “dots” forming in response to TM and overall the signal appears more granular. But whether this is due to trapping of Kar2 into aggregates or other, more functional structures is not known.

*4) Asymmetric retention of mis-folded proteins within the mother compartment is likely to be a time-dependent phenomenon. Perhaps the authors can gain a better insight by performing time-lapse experiments of CPY*-GFP movement between the mother and bud compartments and eventual retention within the mother? Does it seem that accumulation within the mother correlates with the gradual appearance of larger CPY* puncta within the bud? It might be also informative to FRAP CPY*-GFP and/or perform the FRAP of Kar2-GFP at various time-points during TM treatment*.

Indeed, as the reviewers point out and as we explained above, the retention of misfolded proteins is a time dependent phenomenon. With our new CPY*-GFP movies (see Figure 6) we can see that CPY*-GFP accumulates in the mother cell while the bud grows largely free of misfolded proteins. The accumulation of general misfolded proteins is also reported clearly by the slowing down of Kar2-sfGFP dynamics in the mother cells after one hour or two hours Tm treatment (see Figure 1). Finally our new experiments with Sec61-2-GFP shows that this misfolded protein stays in the mother cell while the bud grows (Figure 1 and g, and Figure 7).

The FRAP experiment on CPY*-GFP were done. After FRAP the fluorescence intensity dropped to 37% of the original value. There is then slow recovery and the fluorescence takes 5 minutes to reach 57% of the original value*.* Thus, the aggregate have only slow dynamics.

As explained above, in *bud1Δ* mutant cells we can see the appearance of dots earlier in the bud, before it separates from its mother for CPY*-GFP (see Figure 6). Also wild type buds form aggregates, but only after they have completed mitosis and separated from their mother. Thus, misfolded proteins formed in the mother cell are retained by the barrier in the mother compartment where they may eventually be immobilized further by aggregation. Overall our new data suggest that the formation of large, mainly immobile, and slowly dynamic structures and the retention of smaller misfolded entities by the cER diffusion barrier lead to progressive accumulation of stress in the mother and help the formation of a bud largely free of proteotoxic species. This view is supported by the aging experiments.

*5) The authors address the role of mis-folded protein retention in the longevity of the mother cell. They report that* sur2Δ *and* bud1Δ *mutants had longer and shorter life spans, respectively (*Figure 6*). They conclude that “…the cortical ER diffusion barrier does not participate in the retention of aging factors, …but that the nuclear barrier does.” They then report that a mutant in the ER quality control lectin Yos9 (*yos9Δ*) also resulted in shorter longevity. Significantly,* sur2Δ *and* bud1Δ *mutants in a* yos9Δ *background had ∼normal life-spans, resulting in the authors concluding that retention of mis-folded proteins in the ER of the mother cell accelerate its aging and hence identifies the importance of the cortical ER diffusion barrier. These two sets of experiments appear, especially as written, to be contradictory. Perhaps some re-writing would be helpful*.

We thank the reviewers for pointing out to us this confusing presentation. We edited the text to clarify the issue. Cells lacking ERAD function, and hence experiencing ER stress, are short-lived (*yos9Δ* mutant cells). Furthermore, disruption either of the cortical ER-barrier alone (*yos9Δ bud1Δ* mutant) or both the cortical and the nuclear barriers (*sur2Δ* mutant cells) largely restores the longevity of the mother cells. These data make two points. First, ER stress can act as an aging factor and limit the longevity of the mother cell. Second, opening the cortical barrier is sufficient to ventilate this stress into the mother’s progeny and help it live longer.

Thus, we have asked whether ER stress contributes to limiting the longevity of the mother cell under optimal growth condition, by asking whether disrupting the cortical barrier under such conditions helped the mother cells live longer. This was not the case, although opening both the cortical and the nuclear barriers extends the longevity of the mother. Hence, we conclude that under optimal growth conditions ER stress is not noticeably limiting the lifespan of the mother cells, whereas a nuclear factor probably does.